# OMEGA: Can LLMs Reason Outside the Box in Math? Evaluating Exploratory, Compositional, and Transformative Generalization

**Yiyou Sun[1], Shawn Hu[4], Georgia Zhou[1], Ken Zheng[1], Hannaneh Hajishirzi[2,3], Nouha Dziri[2]\*Dawn Song[1]\***

[1]University of California, Berkeley, [2]Ai2, [3]University of Washington, [4]dmodel.ai

## Abstract

Recent large language models (LLMs) with long Chain-of-Thought reasoning—such as DeepSeek-R1—have achieved impressive results on Olympiad-level mathematics benchmarks. However, they often rely on a narrow set of strategies and struggle with problems that require a novel way of thinking [33]. To systematically investigate these limitations, we introduce OMEGA—**O**ut-of-distribution **M**ath Problems **E**valuation with 3 **G**eneralization **A**xes—a controlled yet diverse benchmark designed to evaluate three axes of out-of-distribution generalization, inspired by Boden's typology of creativity [4]: (1) **Exploratory**—applying known problem-solving skills to more complex instances within the same problem domain; (2) **Compositional**—combining distinct reasoning skills, previously learned in isolation, to solve novel problems that require integrating these skills in new and coherent ways; and (3) **Transformative**—adopting novel, often unconventional strategies by moving beyond familiar approaches to solve problems more effectively. OMEGA consists of programmatically generated training–test pairs derived from templated problem generators across geometry, number theory, algebra, combinatorics, logic, and puzzles, with solutions verified using symbolic, numerical, or graphical methods. We evaluate frontier (or top-tier) LLMs and observe sharp performance degradation as problem complexity increases. Moreover, we fine-tune the Qwen-series models across all generalization settings and observe notable improvements in exploratory generalization, while compositional generalization remains limited and transformative reasoning shows little to no improvement. By isolating and quantifying these fine-grained failures, OMEGA lays the groundwork for advancing LLMs toward genuine mathematical creativity beyond mechanical proficiency. Our code and dataset are available at `https://github.com/sunblaze-ucb/omega`.

## 1 Introduction

Large language models (LLMs) with long Chain-of-Thought (CoT) reasoning—such as DeepSeek-R1 [12], OpenAI-o4 [29], and Claude-Sonnet [35]—have recently achieved impressive results on Olympiad-level mathematics benchmarks, fueling optimism that general-purpose LLMs reasoners may soon rival skilled human problem-solvers. However, recent studies reveal that models trained via Supervised Fine-Tuning (SFT) [33] or Reinforcement Learning (RL) [39] often rely on a limited set of strategies—for instance, such as repeating familiar algebra rules or defaulting to coordinate geometry in diagram problems. And thus, they tend to struggle with particularly challenging problems that require novel insights [33]. Bridging the gap between following learned reasoning patters and demonstrating true mathematical creativity remains a critical open challenge. While fully addressing

---

\*indicates the equal advising role.

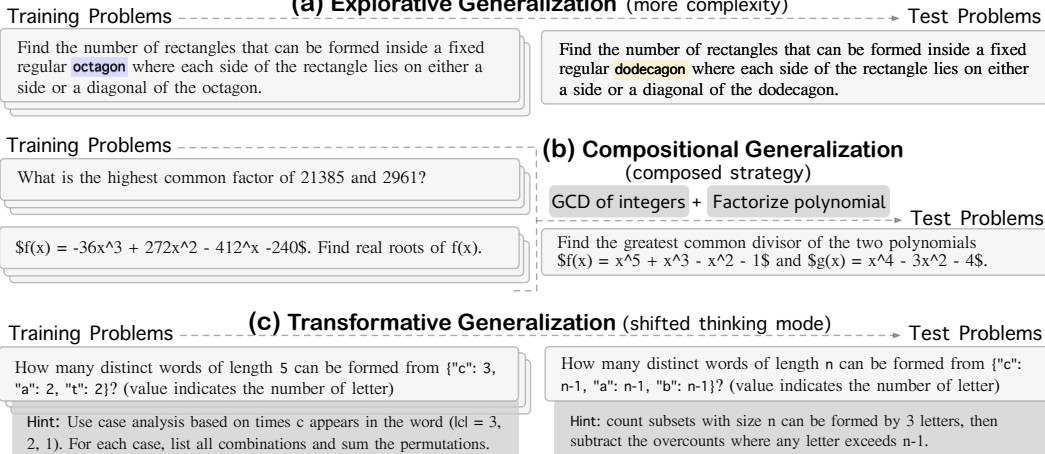

Figure 1: Examples of training-test pairs designed to test distinct generalization capabilities: (a) Explorative Generalization increases complexity within the same frame of thinking (e.g., extending geometric reasoning from an octagon to a dodecagon). (b) Compositional Generalization requires integrating multiple learned strategies (e.g., combining GCD and root-finding for polynomials). (c) Transformative Generalization demands a shift in thinking mode (e.g., from fixed-case enumeration to a "clever" solution that requires thinking in a reverse way).

this gap is an ongoing research effort, our works aims to offer novel insights into the generalization limits of frontier LLMs in mathematical reasoning, cutting through the noise to identify what these models can and cannot do.

Existing math datasets are poorly suited for analyzing math skills that RL models can learn. Large-scale corpora such as *Numina-Math* [20], *Omni-Math* [11], and *DeepMath* [14] blend a large number of math questions in different topics and complexity levels, making it hard to isolate specific reasoning skills behind a model's success or failure. On the other hand, controlled datasets like *GSM-Symbolic* [26], *GSM-PLUS* [22], and *GSM-Infinite* [43] focus on narrow domains, making the diversity of reasoning problems limited. Earlier resources like the *DeepMind Mathematics* suite [2] provide synthetic problems spanning broader topics, but were tailored for earlier-generation models and emphasize elementary tasks (e.g., base conversion), which are far below Olympiad-level complexity. As a result, current benchmarks are either too coarse for causal analysis or insufficiently challenging for modern LLMs. We provide a more detailed comparison in Table 1.

To address this gap, we introduce OMEGA, a controlled, yet diverse benchmark designed to probe three axes of Out-of-Distribution (OOD) generalization, inspired by Boden's typology of creativity [4]. For each axis, we construct matched training-test pairs that isolate a specific reasoning capability (Figure 1) that span 3 dimensions: (1) *Exploratory*—assessing whether models can apply known problem-solving skills to more complex instances within the same problem domain. For example, counting rectangles in an octagon (train) versus a dodecagon (test) (Figure 1a); (2) *Compositional*—evaluating their ability to combine distinct reasoning skills, previously learned in isolation, to solve novel problems that require integrating these skills in new and coherent ways, e.g, finding the GCD of polynomials followed by root-solving (Figure 1b); and (3) *Transformative*—testing whether models can adopt unconventional strategies by moving beyond familiar approaches to solve problems more effectively. For instance, one can replace brute-force enumeration with a subtractive counting method that overcounts and then removes invalid cases (Figure 1c).

OMEGA's test and train problems are constructed using carefully engineered templates that provide precise control over diversity, complexity, and the specific reasoning strategies required for solutions. Our framework employs 40 templated problem generators across six mathematical domains: *arithmetic*, *algebra*, *combinatorics*, *number theory*, *geometry*, and *logic & puzzles*, with complexity levels aligned to Olympiad-level problems. All problems are programmatically generated from problem templates, with answers computed via symbolic, numerical, or graphical methods. Each template encodes a distinct reasoning strategy, enabling systematic generalization studies and the construction of compound problems by combining multiple templates.

Our empirical results reveal three primary findings about current reasoning models: a) **Performance degradation in scaling up complexity.** As mathematical task complexity increases, frontier models'

| Method | Problem Generation | Problem Verification | Overall Complexity | Control w. Cifficulty | Control w. Distribution | Notes |
|--------|--------------------|---------------------|--------------------|-----------------------|-------------------------|-------|
| AIME [3] | Human | Human | High | ✗ | ✗ | 30 Questions per year. |
| GSM8K [6] | Human | Human | Low | ✗ | ✓ | Primitive math-word problems. |
| GSM-Symbolic [26] | Program | Program | Low | ✓ | ✓ | Perturbed math-word problems. |
| GSM-Infinite [43] | Program | Program | Arbitrary | ✓ | ✓ | Infinitely generable math-word problems. |
| MATH500 [30] | Human | Human | Low | ✓ | ✗ | |
| METAMATH [8] | Human/LLM | Human/LLM | Low | ✗ | ✗ | Based on GSM8K/MATH. |
| BigMath [1] | Human | Human/Filters | High | ✗ | ✗ | A mix of many datasets. |
| MathScaleQA [34] | LLM | LLM | Low | ✗ | ✓ | 2M generated datapoints. |
| OpenMath-Instruct [36] | Human/LLM | Human/Code/LLM | Low | ✗ | ✗ | 1.8M solutions to 14K problems from MATH / GSM8K. |
| DeepMath [14] | Human | Human | High | ✓ | ✗ | 103K mathematical problems. |
| **OMEGA** (Ours) | Program | N/A; Correct by Construction | Arbitrarily High | ✓ | ✓ | A controlled dataset for systematic math generalization analysis. |

Table 1: A comparison of various evaluation datasets and the methods used to generate them.

performance deteriorates to near-zero despite substantial inference-time compute. The CoT analysis highlights several key insights: (i) models often discover correct solutions early but expend excessive tokens on verification, leading to inefficient computation, and (ii) models frequently fall into error spirals due to overthinking and self-correction mechanisms, compounding early mistakes and abandoning correct reasoning pathways, (iii) lower accuracy on high-complexity problems can stem from the models' reluctance to perform tedious computations, rather than from arithmetic errors; b) **Generalization of RL exhibits plateau gain.** RL effectively improves model generalization from easy to medium-complexity mathematical problems, especially on familiar (in-domain) tasks, but struggles to achieve significant gains on higher-complexity problems. Performance boosts vary significantly across domains, highlighting the importance of domain-specific knowledge and complexity. c) **Struggle of skill integration and creative reasoning in LLMs.** Unlike humans who fluidly integrate mastered skills, RL models trained on isolated skills struggle at compositional generalization, and models trained conventionally deteriorate on problems necessitating unconventional thinking. These findings underscore crucial gaps between current LLM reasoning capabilities and the flexible, insightful problem-solving characteristic of human mathematicians, particularly in scenarios demanding genuine mathematical creativity beyond mere pattern recognition.

Beyond highlighting current limitations, we hope this study encourages the community to explore smarter scaling solutions rather than brute-force approaches. Although many of the identified failure cases could potentially be patched through targeted data augmentation or synthetic scaffolding, such short-term fixes may obscure deeper, structural weaknesses in model reasoning. Our objective is not only to expose these limitations, but also to inspire strategies that fundamentally equip models with robust, efficient mathematical reasoning capabilities that should address underlying issues that persist beyond simple dataset patches or model scaling.

## 2 OMEGA: Probing the Generalization Limits of LLMs in Math Reasoning

A central goal of mathematical reasoning is not merely to apply memorized procedures but to flexibly adapt, combine, and extend learned strategies. To assess the extent to which LLMs exhibit this capacity, we propose a typology of generalization inspired by *Margaret Boden*'s framework for creativity in cognitive science [4]. Specifically, we define three axes of reasoning generalization—exploratory, compositional, and transformative— to probe the limits of these models on controlled out-of-distribution (OOD) cases that range from easier extensions of seen patterns to harder, more unconventional reasoning problems. Assessing performance along these axes requires fine-grained control over the in-distribution training data.

### 2.1 Problem Construction

Training on a heterogeneous mix of unrelated problems obscures the source of generalization. In contrast, restricting training data to instances drawn from a single template ensures that the model learns a well-scoped strategy.

In our work, all training and test problems problems are generated from carefully designed templates to enable precise control over problem structure, diveristy and required reasoning strategies. To do so, we use 40 *templated problem generators* spanning six mathematical domains: *arithmetic*,

Table 2: Example problem templates across six mathematical domains. For illustration purposes, template content has been shortened. Shaded text indicates programmatically generated variants. Each problem template is associated with a complexity measure $\delta(\theta)$, reflecting task-specific complexity metrics.

| Category | Problem Name | Template Example ($\tau$) with parameter ($\theta$) | Complexity ($\delta(\theta)$) |
|---|---|---|---|
| **Arithmetic** | GCD | What is the greatest common factor of 3450 and 24380 ? | $\log_{10}(\text{answer})$ |
| | Prime Factorization | What is the second-largest prime factor of 519439 ? | $\log_{10}(\text{answer})$ |
| | Mixed Operations | What is the value of (-7920)/1320 - 2/44*4614 ? | number of operations |
| | Matrix Rank | Find the rank of the matrix [[5, -14, 6, -1], [-2, -1, 5, -4], [10, -10, -6, 10], [-19, 1, 3, -31]] | size of the matrix |
| **Algebra** | Linear Equation | Solve 5m = -8k - 345, -3m + 26 + 119 = -898k + 894k for $m$. | number of symbols |
| | Polynomial Roots | Suppose $4160a^3 + 4480a^4 - 585a - \frac{12090}{7}a^2 + \frac{1080}{7} = 0$ what is $a$ (rational number)? | max power |
| | Func Intersection | How many times do the graphs of $f(x) = 2|(-2\sin(\pi x + 2) + 1) - 2| + 3$ and $g(x) = 3|x + 2| - 3$ intersect on $[-10, 10]$? | number of compositions |
| | Func Area | Find the area bounded by $f(x) = 2(-3x + 4)^2 + (-3x + 4) + 3$, $g(x) = 3x - 1$, $x = 1.3$, and $x = 1.7$. | number of compositions |
| **Combinatorics** | Letter Distribution | Distribute {s:3, g:2, j:2} into 3 identical containers holding [3, 2, 2] letters. | number of letters |
| | Pattern Match | Randomly select 3 letters from {o:2, x:3}; expected matches of pattern 'xo+' ? | number of letters |
| | Prob. (No Fixed) | Choose 3 letters from {u:1, f:3, t:2} and shuffle. Probability of no fixed letter positions? | number of letters |
| **Number Theory** | Digit Sum | Let $N$ be the 10th smallest 3-digit integer with digit sum divisible by 6 . Find $N$. | $\log_{10}(\text{answer})$ |
| | Triple Count | How many ordered triples $(a, b, c)$ with $a, b, c \leq 3^2$ satisfy $-2a^3 - 2b^3 + 2c^3 \equiv 0 \pmod{3^2}$ ? | $\log_{10}(\text{answer})$ |
| | Prime Mod | Let $p$ be the smallest prime for which $n^6 + 2 \equiv 0 \pmod{p^5}$ has a solution; find the minimal $n$ for this $p$. | $\log_{10}(\text{answer})$ |
| **Geometry** | Circle | Circle X has center I and radius 8. M has center K and radius 6 and is internally tangent to circle X. Let U be the rotation of point K by angle $7\pi/12$ around point I. Circle D passes through points I, K, and U. What is the radius of circle D? | number of constructions |
| | Rotation | In a regular octagon labeled 1–8 , draw diagonals from 5 to 3 and from 2 to 7 . Rotate the figure 7 steps counterclockwise and overlap it with the original. How many smallest triangular regions are formed? | number of vertices of polygon |
| **Logic & Puzzles** | Grid Blocked | In a 4x4 grid, how many different paths are there from the bottom left (0, 0) to the top right (3, 3), if you can only move right or up at each step, subject to the constraint that you cannot move through the following cells: (3, 1), (2, 3), (0, 1), (2, 1) ? | grid size |

*algebra*, *combinatorics*, *number theory*, *geometry*, and *logic & puzzles*. Example problem templates are illustrated in Table 2. These problems are calibrated at the knowledge level comparable to the American Invitational Mathematics Examination (AIME) [3], with many serving as crucial sub-components in solving Olympiad-level problems. For instance, the `function_intersection` problem type represents an essential building block for questions requiring advanced function analysis.

The selection of problem templates involved several critical considerations:

- **Single-scope with meaningful variations.** Each problem template is designed to focus on a *single-scope* mathematical strategy while allowing for substantial variations. By *single-scope*, we mean that the required solution approach is confined within a well-defined framework, enabling controlled studies of specific reasoning patterns. For instance, instead of combining multiple geometric shapes in a single problem generation template, we isolate problem families on different shapes independently. At the same time, we ensure meaningful variation by designing parameters that fundamentally alter solution trajectories when modified. This contrasts with datasets (numerical perturbation) like *GSM-PLUS* [22], where varying numerical values often preserve the underlying solution path without introducing new reasoning challenges.

- **Programmatic generation and solution validation.** To ensure scalability, both problem instances and their solutions are programmatically generated. This requirement significantly influenced template selection, especially for geometry problems that demand sophisticated procedural generation. We employed diverse computational methods for solution validation: grid search algorithms for `function_intersection` problems, exhaustive enumeration for combinatorial tasks, and computer vision techniques—such as `cv2.approxPolyDP` from `OpenCV`—to accurately count polygons in `rotation` problems.

## 2.2 Training and Evaluation Setup for Generalization

Let $\mathcal{T} = \{\tau\}$ denote a collection of problem *templates*, where each template $\tau$ defines a family of problem instances $\mathcal{P}_\tau = \{\, x_{\tau,\theta} \mid \theta \in \Theta_\tau \,\}$, parameterized by a complexity vector $\theta$ within a parameter space $\Theta_\tau$. We define a scalar *complexity measure* $\delta : \Theta_\tau \to \mathbb{Z}^+$ that ranks problems by increasing complexity. For each generalization axis—such as *exploratory, compositional, or transformative*—we specify a *training* set by selecting a collection of templates along with particular regions of their parameter spaces. Similarly, a distinct set of templates and parameter regions is chosen for *testing* separately, depending on the different generalization test settings. For each generalization category and each math domain, we construct: 1) training data, 2) In-distribution (ID) test data, and 3) OOD test data.

## 2.3 Exploratory Generalization

Exploratory generalization assesses whether a model can *faithfully extend* a single reasoning strategy beyond the range of complexities seen during training. Concretely, the model is exposed to problems drawn from one template $\tau$, all lying within a "low-complexity" regime, and is then evaluated on *harder* instances from the same family. This axis probes robustness: does the model generalizes the same algorithm to higher complexity problems? or does it merely memorize solutions at a fixed complexity level?

**Training and testing data construction**. we define a cutoff threshold $\delta_0$ based on a task-specific complexity measure $\delta$, which determines the maximum complexity level included in training. All problem instances with $\delta \leq \delta_0$ are used for training, while those with $\delta > \delta_0$ are reserved for testing. To ensure the setting remains sufficiently challenging, we select $\delta_0$ such that the base model achieves under 50% accuracy on the training data—reflecting the inherent complexity of these reasoning tasks and leaving room for improvement through fine-tuning. All problem templates introduced in Section 2 are suitable for exploratory generalization experiments, as they encompass scalable reasoning tasks. For each template, we ensure that the complexity scaling aligns with the mathematical intuition of the task, such that increasing $\delta$ genuinely demands more sophisticated reasoning steps.

## 2.4 Compositional Generalization

Compositional generalization probes a model's ability to integrate multiple, distinct reasoning strategies. Unlike explorative generalization, which scales a known method to larger instances, compositional generalization requires a fusion of sub-skills synergistically. Figure 2 illustrates two such cases, where solving the target problem hinges on combining finite-case enumeration with piecewise reasoning or geometric layout analysis with nested-pattern counting. Overall, compositional generalization offers a controlled framework for assessing whether a model can go beyond mastering individual reasoning patterns to dynamically combine them—thereby distinguishing shallow, rote learning from genuine skill integration and true task understanding.

To curate meaningful compositional settings, we enforce the following principles: First, **cohesive skill integration** where the compositional train problems should require true synthesis of multiple reasoning skills rather than superficial concatenation. This ensures that solving the problem depends on the synergistic application of sub-skills, not merely applying them in sequence. Second, **complete skill coverage** where each reasoning skill involved in the composed test task should be independently represented in the training set. This ensures that success on the test reflects the model's ability to compose familiar strategies, rather than rely on exposure to novel ones. And lastly, **nontrivial complexity of train problems** where train problem should be sufficiently challenging so that the model actually learns each sub-skill, making any compositional gains observable. The training problems from our templated inventory remain challenging to the base model, even at low complexity levels (1–2).

**Training and testing data construction**. Our compositional dataset is structured around seven categories (details in Appendix §B.2), each designed to probe specific combinations of reasoning skills. Within each problem family, we identify a core skill and construct corresponding training examples that isolate and reinforce this skill. To evaluate compositional generalization, we then design test problems that require the synergistic application of two distinct skills—such that the solution cannot be obtained by applying each skill naively, but instead demands their true integration. For instance, as illustrated in Figure 2, one problem family focuses on interpreting polygonal geometry, while another targets counting nested patterns; their composition results in a task that requires counting nested structures within polygons. Each setting includes multiple training instances for individual skills and corresponding test instances that assess the model's ability to combine them effectively. Representative examples are provided in Table 7 and Table 8, with additional information in Appendix B.2.

## 2.5   Transformative Generalization

Transformative generalization poses the greatest challenge: it asks whether a model can abandon a familiar but ultimately ineffective strategy in favor of a qualitatively different and more efficient one. These tasks lie outside the scope of mere extension or composition; they require a "jump out of the box"—a creative reframing that circumvents the limitations of standard tactics. To curate meaningful transformative settings, we enforce the following principles: a) **Same problem scope, new insight.** Training and test problems share the same template family (e.g., polynomial-root finding or function-intersection), but test instances are specifically designed so that the familiar tactic either fails or becomes intractably cumbersome; b) **Necessity of reframing.** Solving the test problem must require a novel strategy—such as a symmetry-exploiting substitution or a global geometric argument—rather than exhaustive casework or brute-force enumeration; c) **Nontrivial training tasks.** The training problems themselves remain sufficiently challenging to ensure the model genuinely learns the familiar tactic before being forced to abandon it.

**Training and testing data construction**. Our transformative dataset comprises seven categories (detailed in Appendix §B.3), each specifically designed to evaluate a model's capacity to adopt

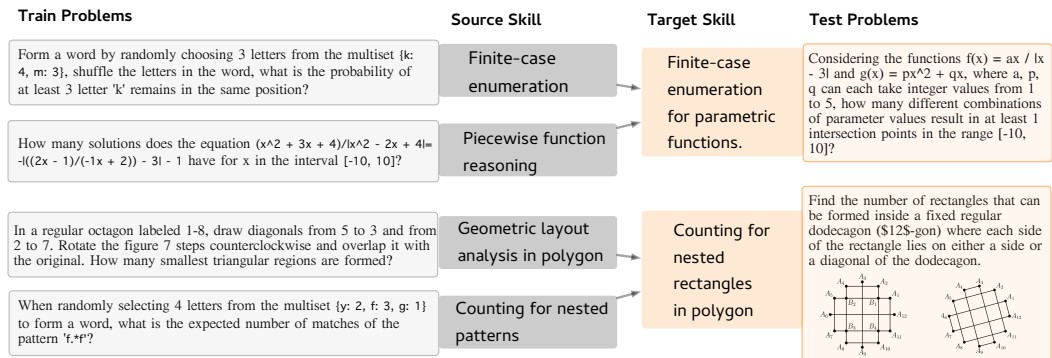

Figure 2: Two examples of compositional generalization in our training/test setup. Each case presents training problems from two separate templates that exercise particular reasoning skills that the model must master, and a test problem that composes the skills. More examples can be found at Appendix B.

Table 3: Illustrative training versus test tasks that probe *Transformative generalization*. Training problems reinforce familiar tactics, but can be over-complicated for test problems where qualitatively different reasoning is required. More examples can be found at Appendix B.

| Problem family | Training regime (familiar tactic) | Transformative test (new tactic required) |
|---|---|---|
| POLYNOMIAL ROOTS | • **Problem.** Solve $f(x) = -36x^3 + 272x^2 - 412x - 240$.
• **Tactic learned.** Apply the Rational Root Theorem (enumerate $p/q$ with $p \mid 240$, $q \mid 36$), test candidates via synthetic division, then factor the cubic. | • **Problem.** Solve $f(x) = x^5 + 10x^3 + 20x - 4$.
• **Needed insight.** Substitute $x = t + \frac{a}{t}$ to exploit symmetry, reduce to a quadratic in $t^2$, then recover $x$. |
| FUNCTION INTERSECTION | • **Problem.** Count intersections of $f(x) = 2|-2\exp(\pi x + 2) + 1| - 2 + 3$ and $g(x) = 3|x + 2| - 3$ on $[-10, 10]$.
• **Tactic learned.** Simplify by sign–case analysis, resolve absolute values, and use periodicity to count intersections. | • **Problem.** With $f(x) = \left\|\|x\| - \frac{1}{2}\right\|$ and $g(x) = \left\|\|x\| - \frac{1}{4}\right\|$, find intersections of $$y = 4g(f(\sin 2\pi x)), \quad x = 4g(f(\cos 3\pi y)).$$
• **Needed insight.** Avoid exhaustive casework; instead, analyze how "up" and "down" graph segments multiply and intersect, using visual symmetry for efficient counting. |

novel problem-solving approaches. Within each category, training problems are generated from the templates described in Section 2. These training tasks can typically be solved using conventional reasoning strategies of moderate complexity, ensuring that the model thoroughly acquires foundational skills. Conversely, the corresponding test problems are intentionally constructed to render these familiar methods ineffective, compelling the model to devise and employ qualitatively distinct solutions. For instance, as illustrated in Table 3, polynomial-root finding tasks in training might be addressed through straightforward factorization, whereas the test scenarios require employing specialized algebraic substitutions to efficiently determine solutions. Similarly, training instances for function-intersection problems might typically involve direct derivative analysis, whereas the test cases demand recognition of underlying geometric properties to bypass computationally intensive algebra. Each transformative category thus pairs multiple training problems that reinforce established techniques with test problems explicitly designed to challenge the model to surpass these traditional approaches and engage in genuine strategic innovation. Additional examples and detailed explanations are available in Appendix §B.3.

# 3 Experiments

## 3.1 Limits of Reasoning Language Models on Increasing Problem Complexity

We evaluate four frontier models—`DeepSeek-R1`, `Claude-3.7-Sonnet`, `OpenAI-o3-mini` and `OpenAI-o4-mini`[2]—across different complexity levels, measuring exact-match accuracy on a held-out set of 100 samples per complexity level. Detailed experimental setup and complexity level descriptions are provided in Appendix C.

**Reasoning LLMs performance degrades with increasing problem complexity** Figure 3 reveals a consistent trend across all models and task types: performance begins near ceiling levels but steadily declines as problem complexity increases. This degradation aligns with the growing number of reasoning steps required, which amplifies the likelihood of error. To justify the evaluation, we provide a complexity analysis in Appendix E, demonstrating that the evaluated problems remain within the models' context length limits. Despite the use of Chain-of-Thought (CoT) traces which enables step-by-step decomposition and self-correction, models still exhibit clear scaling limitations. CoT reasoning remains effective only below a critical complexity threshold, beyond which performance rapidly deteriorates under increased cognitive load. To investigate how CoT reasoning changes under increasing complexity, we analyze the compositional patterns of DeepSeek-R1[3] CoTs in correct and incorrect responses using `O4-mini`. Results are shown in Figure 9.

---

[2]Versions used: Claude (2025-02-19), o3-mini (2025-01-25), o4-mini (2025-04-16).

[3]Analyses of o3 and o4 reasoning traces are not possible since they are hidden per OpenAI policy.

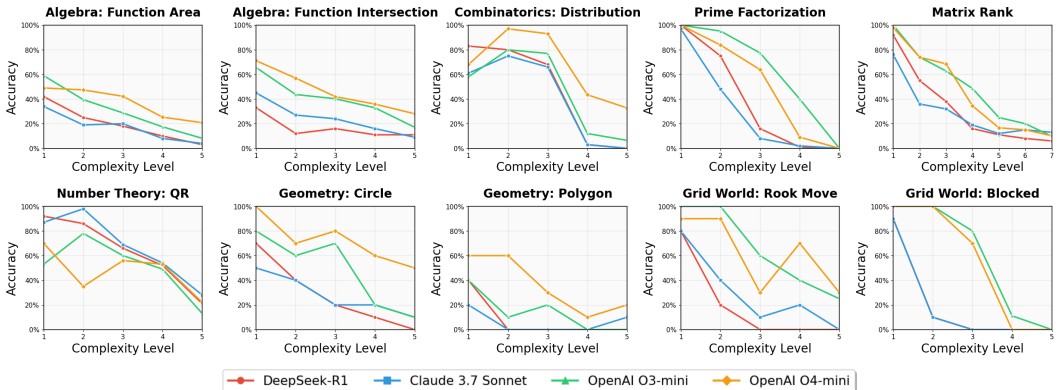

Figure 3: Exact-match accuracy of four top-tier LLMs on OMEGA, plotted against increasing complexity levels. As the complexity increases, performance degrades and goes to zero. We provide complexity analysis to typical problems to ensure they are within the models' output length as detailed in §E.

## 3.2 RL Generalization Experiments

**Experimental Setup.** We evaluate the impact of RL on the generalization capabilities of the base model, *Qwen2.5-7B-Instruct* and *Qwen2.5-Math-7B*[4], across three distinct generalization paradigms: exploratory, compositional, and transformational. For each generalization type, we apply the GRPO algorithm on 1k training problems and evaluate on corresponding in-domain (ID) and out-of-distribution (OOD) test sets. For **exploratory generalization**, we train on problems with restricted complexity levels 1 and 2, then evaluate on: (i) ID problems from the same problem family and complexity range ($\delta \leq \delta_0 = 2$), and (ii) OOD problems from the same problem type but with higher complexity ($\delta > 2$). Regarding **compositional generalization**, for each compositional category $\mathcal{C} = (S_A, S_B)$, we train the model on problems that involve the individual skills $S_A$ and $S_B$ separately, but not their combination, then evaluate on: (i) ID problems testing each skill separately ($P_{S_A}$ and $P_{S_B}$), and (ii) OOD problems requiring the integrated composition of both skills ($P_{S_A \oplus S_B}$), where successful solution demands the synergistic combination rather than sequential application of the individual skills. For **transformational generalization**, we train on problems with conventional solution approaches, then evaluate on: (i) ID problems solvable using familiar methods from the training distribution, and (ii) OOD problems that appear similar to training data but require unconventional solution strategies.

**Can RL Effectively Generalize from Easy to Hard Problems? Strong Early Gains, but Generalization Plateaus with Task Complexity.** Figure 5 shows that RL training on low-complexity problems (levels 1–2) improves generalization to medium-complexity tasks (level 3) across mathematical domains. Gains are consistently larger on in-domain (ID) than out-of-distribution (OOD) examples, indicating that RL primarily reinforces patterns seen during training while still enhancing broader generalization. In the Zebra Logic domain, for instance, RL boosts accuracy from 30% to 91% on ID and 83% on OOD examples—without any supervised fine-tuning—demonstrating that reward-driven

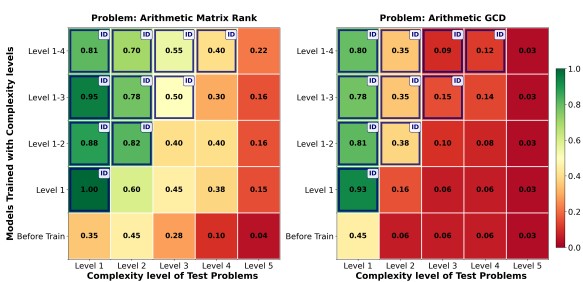

Figure 4: Generalization across complexity levels. Models were trained with data up to a certain complexity level (y-axis) and evaluated on problems from levels 1 to 5 (x-axis). Cells marked 'ID' represent in-distribution evaluations where the test complexity level was included in the training set. Our results show that RL generalizes from easy to hard problems with a plateauing gain.

---

[4]Qwen2.5-Math-7B results follow the same patterns as Qwen2.5-7B-Instruct; please refer to Figure 14 in the Appendix for more details.

exploration alone can yield effective reasoning strategies for combinatorial problems.

However, these gains vary by domain. In geometry, where the base model starts below 15%, RL yields smaller improvements (+31 pp ID, +8 pp OOD). This likely reflects geometry's multimodal complexity (spatial reasoning, diagram understanding, algebraic translation) and limited pretraining exposure. Prior work [40, 27] similarly finds that spatial reasoning requires dedicated data and architectures, suggesting that domain familiarity heavily influences RL effectiveness. Understanding how domain complexity and prior exposure mediate RL gains remains an open question.

We also tested whether broadening RL training to include higher complexities (levels 1–4) improves transfer to harder tasks. As shown in Figure 4, results remain flat: in Arithmetic GCD, accuracy on level 5 stays at 3%, matching the baseline, regardless of whether RL is trained on levels 1–2 or 1–4. RL improves moderately on mid-level tasks (e.g., +4 pp on level 3) but fails to lift performance on the hardest problems. While prior work [24] notes that RL helps most when the base model struggles, our findings reveal no consistent trend across domains—improvements are task-dependent rather than correlated with initial performance.

Overall, RL narrows the ID–OOD gap but does not reliably instill the higher-order reasoning needed to generalize from easy to hard problems.

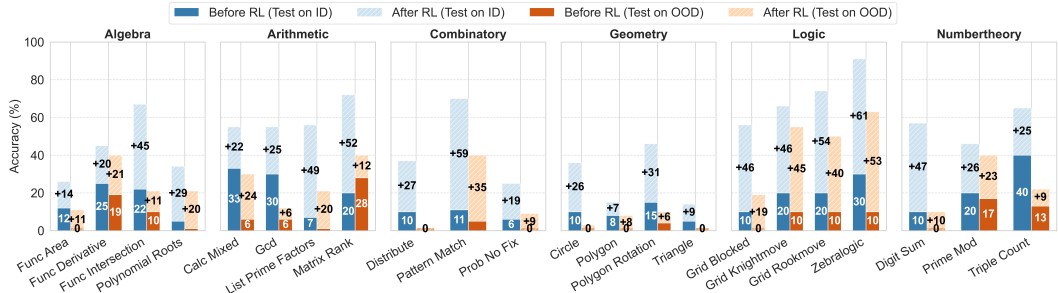

Figure 5: Performance comparison of *Qwen2.5-7B-Instruct* before and after RL on OMEGA under the exploratory generalization setting (Section 2.1). Each problem setting is represented by concatenated bars: In-distribution (ID) accuracy (blue) and Out-of-distribution (OOD) accuracy (orange). RL yields strong improvements across most domains on in-distribution tasks; however, gains on out-of-distribution tasks are typically lower and more variable, highlighting the limits of generalization from seen distributions.

### 3.2.1 Can RL Learn to Compose Math Skills into Integrated Solutions? Strong Performance on Isolated Skills, but Limited Compositional Generalization

We test whether RL enables models to combine distinct reasoning skills learned separately into novel compositions. For each compositional category $C_i = (S_{a_i}, S_{b_i})$, RL trains the model on problems requiring each skill in isolation, excluding any joint occurrences. The OOD evaluation then measures whether the model can solve problems requiring $S_{a_i} \oplus S_{b_i}$—a direct test of emergent compositional reasoning.

Figure 6 shows that RL reliably strengthens individual skills (often >69% accuracy on $S_{a_i}$ and $S_{b_i}$). For example, polygon rotation accuracy rises from 13% to 69%, and pattern matching from 6% to 16%. However, the magnitude of improvement varies by skill, suggesting that some are inherently easier to reinforce. In contrast, models show little to no improvement on compositional tasks. Even

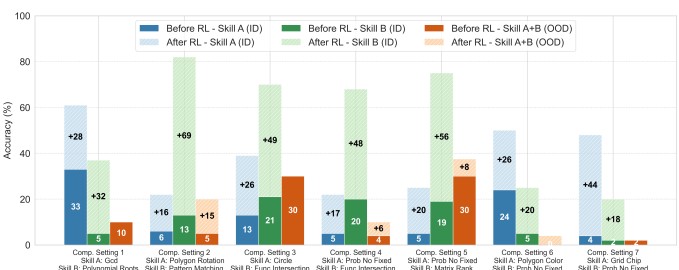

Figure 6: Performance comparison of *Qwen2.5-7B-Instruct* on OMEGA under the compositional generalization setting. The model's ability to integrate reasoning strategies from two problem families is assessed. For each setting, accuracies are reported on the individual in-distribution problem families (Skill A & Skill B) and their compositional problems. Results are shown before and after RL.

after RL, combining skills (e.g., GCD + polynomial roots) yields negligible gains (typically <6 pp). This indicates that RL mainly reinforces task-specific patterns rather than fostering flexible reasoning that can generalize to novel compositions—unlike human reasoning, which naturally recombines learned procedures to solve new problems.

Ablation studies (Tables 11–12) further confirm that compositional gains depend on the conceptual alignment of the underlying skills. Original pairings yield moderate improvements (+7.5–15 pp), while altering one or both components sharply reduces or negates these gains. Thus, RL supports limited compositional generalization, effective only when component skills are closely related and jointly reinforced during training.

### 3.2.2 Can RL Go Beyond Familiar Skills to Discover New Reasoning Abilities? Learns Familiar Strategies, but Struggles with Unconventional Solution Paths

Figure 7 presents the most challenging test of reasoning flexibility, requiring models to abandon training-observed strategies and discover entirely new solution approaches. The base model performs modestly across most settings, with accuracy typically below 20%. RL training provides substantial benefits on in-domain examples where the solution approach is familiar from training data, and this aligns with our previous generalization tasks (e.g., +56% on matrix rank).

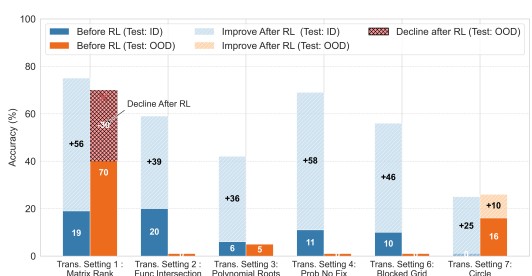

Figure 7: Performance comparison of *Qwen2.5-7B-Instruct* on OMEGA under the transformational generalization setting. The model's ability to adopt qualitatively new reasoning strategies is evaluated.

However, performance on OOD transformational problems remains low after RL, often 0%. The only notable improvement appears in Setting 7 (+10 pp); however, upon examining the model's trajectories, we observe that it continues to rely on naive solutions, succeeding only on some simple variants of the transformative problems where conventional approaches still apply. Nonetheless, the result highlights both the strengths and limits of RL: it offers meaningful gains when a familiar structure exists but struggles to induce genuinely novel reasoning strategies without prior exposure. This suggests that RL training alone could be insufficient for discovering novel reasoning paradigms, and that such transformational capabilities may require explicit exposure to diverse problem-solving strategies during base model training or supervised fine-tuning. Notably, in the matrix rank setting where the base model achieved decent OOD performance (70%), further RL training actually led to performance deterioration, dropping 30 percentage points. This decline indicates that RL optimization can sometimes reinforce suboptimal patterns learned during training rather than promoting exploration of alternative approaches.

## 4 Discussion & Conclusion

We have presented OMEGA, a controlled benchmark designed to isolate and evaluate three axes of out-of-distribution generalization in mathematical reasoning: explorative, compositional, and transformative. We provide a detailed discussion with related works in Appendix § A. In this work, by generating matched train–test pairs from template-driven problem families, our framework enables precise analysis of reasoning behaviors and supports infinite-scale, reproducible synthesis. Our empirical study yields three key insights. First, RL fine-tuning delivers substantial gains on both in-distribution and explorative generalization, boosting accuracy on harder instances within known problem domains. Second, despite these improvements, RL's impact on compositional tasks remains modest: models still struggle to integrate multiple learned strategies coherently. Third, RL struggles to induce genuinely new reasoning patterns, showing negligible progress on transformative generalization that requires shifting to novel solution paradigms. These findings underscore a fundamental limitation: while RL can amplify the breadth and depth of problems that LLMs solve, they do not by themselves foster the creative leaps needed for true transformational reasoning. By diagnosing where and why current LLMs fail to generalize creatively, OMEGA lays the groundwork for next-generation reasoners that can not only interpolate but also innovate—moving us closer to human-level mathematical problem-solving.

## Acknowledgements

This work was supported in part by the National Science Foundation under the ACTION program. Computational resources were provided by the Allen Institute for AI (AI2) cluster. We thank all collaborators and reviewers for their valuable feedback and contributions.

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

# A  Related Work

**OOD Generalization and Compositional Abilities of LLMs.** Generalization to out-of-distribution (OOD) data remains a fundamental challenge in large language models (LLMs) and machine learning more broadly, with far-reaching implications for tasks such as mathematical reasoning, physical modeling, and financial forecasting [10, 17, 21, 23, 37, 38]. In practice, many key questions about model performance reduce to whether models can effectively handle test distributions that differ from their training data. Compositional generalization—models' ability to systematically combine learned skills—has also been a long-standing focus in language research [5, 9, 18, 25, 15, 16, 32]. Much of this work has relied on controlled testbeds involving rule-based languages such as SQL or synthetically generated tasks. More recently, [42] extended this line of inquiry to natural language skills, while [7] examined whether LLMs can acquire compositional generalization through tasks like integer multiplication and dynamic programming. Building on this foundation, OMEGA offers a comprehensive benchmark for assessing compositional generalization in mathematical reasoning, spanning a broad range of problem types and solution strategies.

**Benchmarking LLMs' Mathematical Abilities.** The most common way to evaluate an LLM's math ability is by reporting accuracy on a large collection of questions. They are typically created in a few ways: by hiring humans to write problems (e.g., GSM8K [28], MinervaMath [19]), which allows control over topic and complexity but is costly; by collecting or adapting existing exam questions (e.g., AIME [3], OlympiadBench [13], GaoKao [41]), which ensures quality but limits scale and diversity; or by scraping exam corpora and filtering them with human (e.g., NuminaMath [20], BigMath [1]); or LLM-based verification (e.g., MathScale [34]). Another approach is to generate problems using LLMs with correctness constraints (e.g., MetaMathQA [8], OpenMathInstruct-1 [36]). Some works also modify existing datasets for specific goals, like GSM-Plus [22], GSM-Symbolic [26] and GSM-Infinite [43]. Other typical datasets include Math500 [30], AIME [3], MinervaMath [19], and OlympiadBench [13]. The detailed comparison of popular math benchmarks is in Table 1. In the recent study [31], evaluations on four synthetic puzzles indicate that LLMs encounter distinct reasoning boundaries as problem complexity escalates. These findings align with our observations presented in Section 3.1, where we examine a broader set of mathematical categories across multiple problem families. Beyond evaluating the performance limits of frontier models, our work further investigates the underexplored boundaries of reasoning generalization in LLMs–specifically through explorative, compositional, and transformative perspectives.

# B  Dataset Details

## B.1  Details of Problem Families

To provide full transparency on our templated generators, we include three comprehensive tables in the appendix. Table 4 lists all arithmetic and algebra templates (e.g., linear equations, polynomial roots, function operations), alongside their complexity measures across five calibration levels. Table 5 details the combinatorics and number-theory generators with corresponding size or range metrics at each level. Finally, Table 6 presents our logic & puzzles and geometry templates, again annotated with statement counts or grid sizes for the five levels. Together, these tables document the full set of 41 problem families used in MathOOD, illustrating how each template is systematically calibrated to enforce controlled, domain-specific reasoning strategies.

Table 4: Problem families (arithmetic and algebra) with sample problems and complexity measures across five levels.

| Problem Family Alias | Sample Problem Statement | Complexity Measure | Lv1 | Lv2 | Lv3 | Lv4 | Lv5 |
|---|---|---|---|---|---|---|---|
| algebra/linear_equation | Solve $3n - 4t + 1012801 = 1012843$, $-3n + 66 = 4t$ | Symbol number | 2 | 3 | 4 | 5 | 6 |
| algebra/polynomial_roots | Express the second largest root of $-\frac{147407}{8}m^3 - \frac{19331}{4}m^2 + \frac{1053}{8}m + \frac{117}{2} = 0$ as $n/m$ where $\gcd(n,m) = 1$. | Degree | 3 | 4 | 5 | 6 | 7 |
| algebra/func_integration | Compute the indefinite integral for $f(x) = 2(x-5)^2 - 4(x-5) + 3$. | Composed function number | 2 | 3 | 4 | 5 | 6 |
| algebra/func_area | Determine the area enclosed by $f(x) = \frac{3(-e^{-x}-2)-1}{-3(-e^{-x}-2)-3}$, $\quad g(x) = -3|x+1|+3$ | Composed function number | 2 | 3 | 4 | 5 | 6 |
| algebra/func_derivative | Number of maximal connected intervals in $[-10, 10]$ where $f(x) = -4\left(-2\sin(\pi x - 2) + 2\right) + 5$ is increasing. | Composed function number | 2 | 3 | 4 | 5 | 6 |
| algebra/func_ext_coords | Average of all $x$-coordinates of local minima of $f(x) = \frac{-3(-2\sin(\pi x - 2)+2)+2}{2(-2\sin(\pi x - 2)+2)+1}$. | Composed function number | 2 | 3 | 4 | 5 | 6 |
| algebra/func_extrema | Number of local maxima of $f(x) = 2\cos\left(3\pi(|x+1|+3)+3\right) - 1$ in $[-10, 10]$. | Composed function number | 2 | 3 | 4 | 5 | 6 |
| algebra/func_intsct_coords | Integer value (rounded) at which $f(x) = x - 5$, $\quad g(x) = -2|x| - 1$ intersect in $[-10, 10]$. | Composed function number | 2 | 3 | 4 | 5 | 6 |
| algebra/func_intersection | Number of intersections of $f(x) = -3\cos(2\pi(2|x+2|+2)+3)+1$, $\quad g(x) = 4x - 3$ in $[-5, 5]$. | Composed function number | 2 | 3 | 4 | 5 | 6 |
| algebra/func_zeros | Number of $x$-intercepts of $f(x) = 3\cos\left(\pi(-3|x-2|+1)-3\right) + 3$. | Composed function number | 2 | 3 | 4 | 5 | 6 |
| arithmetic/gcd | What is the greatest common divisor of 1290 and 64715? | Digit length | [4,7] | [10,12] | [15,20] | [20,25] | [25,30] |
| arithmetic/calc_mixed | Evaluate $-2-((-9)/7+((-1632)/119-0))$. | Operation length | [4,9] | [10,14] | [14,16] | [16,20] | [20,25] |
| arithmetic/list_prime | Find the second-largest prime factor of 62033. | Max answer | 25 | 100 | 200 | 400 | 800 |
| arithmetic/determinant | Determine $\det(A)$. | Row | 3 | 4 | 6 | 7 | 9 |
| arithmetic/eigenvalues | Find eigenvalues of $A$ and report the largest (by absolute value). | Row | 3 | 4 | 6 | 7 | 9 |
| arithmetic/inverse | Invert $\frac{1}{60}A$ and sum all entries of the inverse. | Row | 3 | 4 | 6 | 7 | 9 |
| arithmetic/multiplication | Entry $(2,1)$ of the product of given matrices $A$ and $B$. | Row | 3 | 4 | 6 | 7 | 9 |
| arithmetic/power | Sum of all entries of $A^2$. | Row | 3 | 4 | 6 | 7 | 9 |
| arithmetic/rank | Rank of the matrix $A$. | Row | 3 | 4 | 6 | 7 | 9 |
| arithmetic/svd | Rounded largest singular value of $A$ in its SVD. | Row | 3 | 4 | 6 | 7 | 9 |

Table 5: Problem families (combinatory and number theory) with sample problems and complexity measures across five levels.

| Problem Alias Family | Sample Problem Statement (simplified) | Complexity Measure | Lv1 | Lv2 | Lv3 | Lv4 | Lv5 |
|---|---|---|---|---|---|---|---|
| combinatory/distribute | Divide the letters from {'m' : 2, 'p' : 2, 't' : 2} into 3 distinctively labeled boxes with sizes $[2, 1, 3]$. How many ways? | Total letters | [4,6] | [6,8] | [9,10] | [11,11] | [12,12] |
| combinatory/pattern_match | Form a word by randomly choosing 4 letters from the multiset {'h' : 6, 'u' : 3}. What is the expected number of occurrences of h.*h? | Total letters | [4,6] | [6,8] | [9,10] | [11,12] | [13,14] |
| combinatory/prob_gt_n_fix | What is the probability that, when forming a 4-letter word from {'h' : 2, 'r' : 3, 'q' : 3} and shuffling it, at least one 'r' remains in its original position? | Total letters | [4,6] | [6,8] | [8,9] | [10,11] | [11,12] |
| combinatory/prob_eq_n_fix | What is the probability that, when forming a 2-letter word from {'m' : 2, 'r' : 1, 'o' : 1} and shuffling it, exactly one 'r' remains in its original position? | Total letters | [4,6] | [6,8] | [8,9] | [10,11] | [11,12] |
| combinatory/prob_no_fix | What is the probability that, when forming a 4-letter word from {'b' : 4, 'i' : 2, 'u' : 2} and shuffling it, no letter remains in its original position? | Total letters | [4,6] | [6,8] | [8,9] | [10,11] | [11,12] |
| combinatory/prob_no_letter | What is the probability that, when forming a 4-letter word from {'r' : 3, 'x' : 3, 'n' : 2} and shuffling it, no 'x' occupies any of its original positions? | Total letters | [4,6] | [6,8] | [8,9] | [10,11] | [11,12] |
| numbertheory/digit_sum | Let $N$ be the greatest 4-digit integer such that both $N$ and its digit-reverse are divisible by 9. What is the digit sum of $N$? | Digit count | 2 | 3 | 4 | 5 | 6 |
| numbertheory/triple_count | Let $N$ be the number of ordered pairs $(a, b)$ with $a, b \leq 2^4$ such that $a^2 + b^2$ is a multiple of $2^2$. What is $N$? | Max answer | 10 | 50 | 100 | 200 | 500 |
| numbertheory/prime_mod | Let p be the least prime number for which there exists a positive integer n such that $n^3 + (2)$ is divisible by $p^4$. Find the least positive integer m such that $m^3 + (2)$ is divisible by $p^4$. | Digit count | 2 | 3 | 4 | 5 | 6 |

Table 6: Problem families (logic and geometry) with sample problems and complexity measures across five levels.

| Problem Family Alias | Sample Problem Statement (simplified) | Complexity Measure | Lv1 | Lv2 | Lv3 | Lv4 | Lv5 |
|---|---|---|---|---|---|---|---|
| logic/blocked_grid | In a $3 \times 6$ grid, how many paths from $(0,0)$ to $(2,5)$, moving only right or up, if cells $(0,4),(1,3),(2,0)$ are forbidden? | Grid size | [5,10] | [10,20] | [20,30] | [30,50] | [50,70] |
| logic/grid_rook | In a $3 \times 6$ grid, minimal rook-like moves (any number right or up) from $(0,0)$ to $(2,5)$, avoiding $(1,1),(1,0),(1,3),(2,0)$? | Grid size | [5,10] | [10,20] | [20,30] | [30,50] | [50,70] |
| logic/grid_knight | On an $8 \times 9$ grid, minimal knight-like moves (5 by 1 leaps) from $(0,0)$ to $(7,5)$? | Grid size | [5,10] | [10,20] | [20,30] | [30,50] | [50,70] |
| logic/zebralogic | Two houses numbered 1–2 each with unique person (Arnold, Eric), birthday (april, sept), mother (Aniya, Holly). Clues: Eric is left of Holly's child; April birthday in house 1. Which choice index? | max(# of attributes, # of people) | 2 | 3 | 4 | 5 | 6 |
| logic/grid_chip | In a $5 \times 5$ grid, chips black/white satisfy row/column uniformity and maximality; given colours at $(3,4),(2,0),(4,3),(1,1),(2,2),(0,3)$. How many chips placed? | Grid size | 4 | 5 | 6 | 7 | 8 |
| geometry/basic | $DS = 10$. $P$ is midpoint of $DS$. Rotate $S$ by $7\pi/12$ about $P$ to $X$. Reflect $X$ over $D$ to $Z$; reflect $D$ over $Z$ to $L$. $B$ is midpoint of $PZ$; $F$ is bisector of $\angle SPL$; reflect $S$ over $F$ to $T$. Find $|BT|$. | Statement number | 10 | 15 | 20 | 25 | 30 |
| geometry/polygon_chords | For a 6-gon with specified diagonals drawn (2–6,1–4,3–6,5–2,6–4,4–2,3–1), how many pairs of diagonals are perpendicular? | # of diagonals | [6,7] | [8,9] | [10,11] | [12,13] | [14,15] |
| geometry/circle | Circle center $C$, radius 7. $G$ on circle; $L$ midpoint of $GC$; $X$ midpoint of $LC$; $I$ midpoint of $LX$; $F$ is reflection of $G$ across $C$. Find $|IF|$. | Statement number | 10 | 15 | 20 | 25 | 30 |
| geometry/polygon_general | Square $ABCD$ center $T$, circumradius 7. Reflect $T$ across $B$ to $G$. $O$ midpoint of $DG$; $Z$ midpoint of $TA$. Find $|OZ|$. | Statement number | 10 | 15 | 20 | 25 | 30 |
| geometry/triangle | $XT = 6$. Rotate $T$ by $5\pi/6$ about $X$ to $O$. Reflect $O$ across $XT$ to $V$. $D$ is incenter of $\triangle TOX$; $E$ midpoint of $XV$. Find $|DE|$. | Statement number | 10 | 15 | 20 | 25 | 30 |
| geometry/rotation | In a 10-gon, draw diagonals 5–9 and 8–6, then rotate setup 5 vertices CCW and superimpose. Count smallest polygons formed. | Diagonal number | 2 | 3 | 4 | 5 | 6 |
| geometry/polygon_color | A 6-gon vertices colored B,B,R,B,B,B in order. By rotating, what is the maximum blue vertices landing on originally red positions? | $n$ of $n$-gon | [6,7] | [8,9] | [10,11] | [12,13] | [14,15] |

## B.2 Details of Compositional Generalization Problems

Compositional generalization evaluates a model's ability to integrate multiple, distinct reasoning strategies. In contrast to exploratory generalization—which focuses on scaling a single known method to larger instances—compositional generalization requires the synergistic fusion of sub-strategies to solve more complex problems. By the submission deadline, we provide 7 distinct settings to assess compositional performance. Setting 1, illustrated in Figure 1, combines GCD and polynomial root problems. Detailed examples and explanations for the remaining six settings are provided in Table 7 and Table 8.

Table 7: Examples (part 1) of training and test tasks that probe *Compositional generalization* ability of LLM.

| Problem family | Training regime (familiar tactic) | Compositional test (combined tactic required) |
|---|---|---|
| COMP. SETTING 2: GEOMETRY/ROTATION + COMBINATORY/PATTERN_MATCH | • **Example Problem from Domain A.** Suppose you have a 9-gon, with vertices numbered 1 through 9 in counterclockwise order. Draw the diagonal from vertex 6 to vertex 4, from vertex 1 to vertex 6, and from vertex 3 to vertex 5. Then, rotate the entire setup, including the constructed diagonals, 8 vertices counterclockwise (so that vertex 1 ends up where vertex 9 was), and superimpose it on the original (so that the resulting diagram contains both the original diagonals and the rotated versions of the diagonals). The original 9-gon will be partitioned into a collection of smaller polygons. How many such polygons will there be? 
• **Example Problem from Domain B.** Form a word by randomly choosing 3 letters from the multiset {y: 2, v: 1, p: 4, z: 4}. What is the expected number of occurrences of the pattern 'p.*p' in each word? | • **Composed Problem.** Find the number of rectangles that can be formed inside a fixed regular 12-gon where each side of the rectangle lies on either a side or a diagonal of the 12-gon. Note that it is possible for a rectangle to be contained within another rectangle, and that the rectangles may not extend beyond the boundaries of the 12-gon. 
• **Decomposition.** After observing the rotational symmetries of the 12-gon and "visualizing" the problem, define the conditions necessary for lines parallel/perpendicular to a specific orientation to form a rectangle. Since a rectangle divided along an line parallel to its sides forms more rectangles, finding the number of total rectangles in such a structure is a combinatorial problem isomorphic to the string problem. |
| COMP. SETTING 3: GEOMETRY/CIRCLE + ALGEBRA/FUNC_INTERSECTION | • **Example Problem from Domain A.** Circle center $C$, radius 7. $G$ on circle; $L$ midpoint of $GC$; $X$ midpoint of $LC$; $I$ midpoint of $LX$; $F$ is reflection of $G$ across $C$. Find $|IF|$. 
• **Example Problem from Domain B.** Find the number of intersections of $f(x) = -3\cos(2\pi(2|x+2|+2)+3)+1$, $g(x) = 4x-3$ in $[-5,5]$. | • **Composed Problem.** A circle with radius 4 is moving on the coordinate plane such that its center moves along the curve $P(t) = \langle t, t^2 \rangle$ starting at t=0. Find the first value of t for which the circle lies tangent to the x-axis. 
• **Decomposition.** Observe that it is sufficient to find a value of t for which the circle's center has a y-coordinate of 4, which reduces to a pure "equation solving" problem. [5] |
| COMP. SETTING 4: COMBINATORY/PROB_NO_FIX + ALGEBRA/FUNC_INTERSECTION | • **Example Problem from Domain A.** What is the probability that, when forming a 4-letter word from {'b' : 4, 'i' : 2, 'u' : 2} and shuffling it, no letter remains in its original position? 
• **Example Problem from Domain B.** Find the number of intersections of $f(x) = -3\cos(2\pi(2|x+2|+2)+3)+1$, $g(x) = 4x-3$ in $[-5,5]$. | • **Composed Problem.** Considering the functions $f(x) = a\sin(b\pi x)$ and $g(x) = p\sin(\pi qx)$, where a, b, p, q can each take integer values from 1 to 5, how many different combinations of parameter values result in at least 7 intersection points in the range [-10, 10]? 
• **Decomposition.** The composed problem requires integrating symbolic reasoning over parameterized trigonometric functions (from Domain B) with combinatorial generalization over multiple configurations (related to Domain A). |

Table 8: Examples (part 2) of training and test tasks that probe *Compositional generalization* ability of LLM.

| Problem family | Training regime (familiar tactic) | Compositional test (combined tactic required) |
|---|---|---|
| COMP. SETTING 5: ARITHMETIC/MATRIX_RANK + COMBINATORY/PROB_NO_FIX | • **Example Problem from Domain A.** Compute the rank of the given 4x4 matrix: ... 
 • **Example Problem from Domain B.** What is the probability of such event happening: Form a word by randomly choosing 4 letters from the multiset {j: 4, d: 2, p: 2}, shuffle the letters in the word, what is the probability of exact 1 letter 'p' remains in the same position? | • **Composed Problem.** Consider the matrix $M = \begin{bmatrix} a & b & c \\ 1 & a & b \\ 2 & 1 & a \end{bmatrix}$ where a, b, and c are integers between 3 and 10, inclusive. How many different combinations of (a, b, c) result in a matrix with rank exactly 3 
 • **Decomposition.** The composed problem requires integrating linear algebra reasoning (matrix rank determination) (from Domain B) with combinatorial generalization over multiple configurations (related to Domain A). |
| COMP. SETTING 6: GEOMETRY/POLYGON_COLOR + COMBINATORY/PROB_NO_FIX | • **Example Problem from Domain A.** A 6-gon is colored so that in clockwise order, the vertices are colored as follows: vertex 0 is blue, vertex 1 is blue, vertex 2 is red, vertex 3 is blue, vertex 4 is blue, vertex 5 is blue. What is the maximum number of blue vertices that can be made to occupy a position where there were originally red vertices by rotating the 6-gon? 
 • **Example Problem from Domain B.** What is the probability of such event happening: Form a word by randomly choosing 4 letters from the multiset {j: 4, d: 2, p: 2}, shuffle the letters in the word, what is the probability of exact 1 letter 'p' remains in the same position? | • **Composed Problem.** Each vertex of a regular octagon is independently colored either red or blue with equal probability. The probability that the octagon can then be rotated so that all of the blue vertices end up at positions where there were originally red vertices is $\frac{m}{n}$, where $m$ and $n$ are relatively prime positive integers. What is $m + n$? 
 • **Decomposition.** The problem is fundamentally about finding the number of cases satisfying a constraint. The first subproblem tests understanding of the constraint (and the required spatial reasoning). The second subproblem tests the ability to enumerate cases. |
| COMP. SETTING 7: LOGIC/GRID_CHIP + COMBINATORY/PROB_NO_FIX | • **Example Problem from Domain A.** Chips, colored either black or white, are placed in the 25 unit cells of a 5x5 grid such that: a) each cell contains at most one chip, b) all chips in the same row and all chips in the same column have the same colour, c) any additional chip placed on the grid would violate one or more of the previous two conditions. Furthermore, we have the following constraints (with the cells 0-indexed): cell (3, 4) is black, cell (2, 0) is white, cell (4, 3) is black, cell (1, 1) is white, cell (2, 2) is white, cell (0, 3) is black. How many chips are placed on the grid? 
 • **Example Problem from Domain B.** What is the probability of such event happening: Form a word by randomly choosing 4 letters from the multiset {j: 4, d: 2, p: 2}, shuffle the letters in the word, what is the probability of exact 1 letter 'p' remains in the same position? | • **Composed Problem.** There is a collection of 25 indistinguishable white chips and 25 indistinguishable black chips. Find the number of ways to place some of these chips in the 25 unit cells of a $5 \times 5$ grid such that: 
   – each cell contains at most one chip 
   – all chips in the same row and all chips in the same column have the same colour 
   – any additional chip placed on the grid would violate one or more of the previous two conditions. 
 • **Decomposition.** The problem asks to find the number of possible arrangements subject to the named constraints. The first subproblem tests understanding of constraints in a very similar setting. The second subproblem tests the ability to compute the number of cases fitting a particular constraint. |

## B.3 Details of Transformative Generalization Problems.

Transformative generalization presents the greatest challenge: it tests whether a model can discard a familiar yet ineffective strategy in favor of a qualitatively different and more efficient one. These tasks go beyond simple extension or composition, requiring a "jump out of the box"—a creative reframing or redescription that bypasses the limitations of standard reasoning tactics. By the submission deadline, we include 7 distinct settings to evaluate transformative generalization. Setting 2 (algebra/function_intersection) and Setting 3 (algebra/polynomial_root) are illustrated in Table 3, while Setting 4 (combinatory/prob_no_fix) is visualized in Figure 1. Detailed examples and explanations for the remaining settings are provided in Table 9 and Table 10.

Table 9: Examples of training and test tasks that probe *Transformative generalization* (part 1)

| Problem family | Training regime (familiar tactic) | Transformative test (new tactic required) |
|---|---|---|
| TRANSFORMATIVE SETTING 1: MATRIX_RANK | • **Problem.** What is the rank of the matrix: $\begin{bmatrix} -4 & -16 & -8 & 7 \\ 9 & 17 & 6 & -14 \\ 4 & 10 & 0 & -10 \\ 7 & 6 & -2 & -12 \end{bmatrix}$ item **Tactic learned.** Use Gaussian elimination to reduce the matrix to row-echelon form and count the number of nonzero pivot rows. | • **Problem.** Let $E_n$ be $n \times n$, $e_{ij} = \begin{cases} 1 & \text{if } i+j \text{ is even} \\ 0 & \text{if } i+j \text{ is odd} \end{cases}$. Find $\text{rank}(E_n)$. 
 • **Needed insight.** Observe that $$E_n = \tfrac{1}{2}\big(\mathbf{1}\mathbf{1}^T + [(-1)^i]_i\,[(-1)^j]_j^T\big),$$ i.e. a sum of two outer products (each rank 1), so $\text{rank}(E_n) = 2$ for $n \geq 2$ (and 1 if $n = 1$). |
| TRANSFORMATIVE SETTING 5: FUNC_INTEGRATION | • **Problem.** What is the symbolic integration of the function $$f(x) = 4\big(-1(5x^2 + 5x - 2) + 4\big) - 3?$$ • **Tactic learned.** First expand and simplify the algebraic expression to a polynomial, then apply the power-rule integration term by term. | • **Problem.** Evaluate the indefinite integral $$\int (1+x+x^2+x^3+x^4)\,(1-x+x^2-x^3+x^4)\,dx.$$ • **Needed insight.** Observe that multiplying the two quintic sums collapses all odd-power terms, yielding the even-power polynomial $x^8 + x^6 + x^4+x^2+1$, which can then be integrated directly by the power rule. |

Table 10: Examples of training and test tasks that probe *Transformative generalization* (part 2).

| Problem family | Training regime (familiar tactic) | Transformative test (new tactic required) |
|---|---|---|
| TRANSFORMATIVE SETTING 6: LOGIC/BLOCKED_GRID | • **Problem.** In a 6x6 grid, how many different paths are there from the bottom left (0, 0) to the top right (5, 5), if you can only move right or up at each step, subject to the constraint that you cannot move through the following cells: (3, 3), (2, 1), (3, 4), (3, 1), (0, 5), (5, 0), (2, 0), (0, 4), (2, 5) 
 • **Tactic learned.** Among possible strategies, plot the cells on the grid, and categorize paths according to whether they pass above or below a fixed cell. Use combinatorial formulas to easily find the number of paths in each category. For smaller problems, use brute-force search. | • **Problem.** In a 10x10 grid, how many different paths are there from the bottom left (0, 0) to the top right (9, 9), if you can only move right or up at each step, subject to the constraint that you cannot move through the following cells: (2, 0), (2, 1), (2, 2), (2, 3), (2, 4), (2, 5), (2, 6), (2, 7), (2, 8)? 
 • **Needed insight.** There is a wall, which vastly simplifies the analysis. The only variation among viable paths is at which "vertical" index we first choose to move right, so there are 10 options. |
| TRANSFORMATIVE SETTING 7: GEOMETRY/CIRCLE | • **Problem.** Let C be the circle with center V and radius 6. Point K is on circle C. Let I be the midpoint of segment KV. Point M is the midpoint of segment IK. Let L be the midpoint of segment IM. What is the distance between points L and I? 
 • **Tactic learned.** Construct circles, lines, and perpendicular bisectors; find distances between relevant points in the plane using coordinate geometry. | • **Problem.** Let circle $C_1$ be positioned in the coordinate plane with a radius of 1. Draw its horizontal diameter and call its endpoints $A_1$ and $B_1$. Draw its vertical diameter and call the higher endpoint $D_1$. Then, let circle $C_2$ be the circle centered at $D_1$ that passes through $A_1$ and $B_1$. Likewise, draw its horizontal diameter and call its endpoints $A_1$ and $B_1$, and draw its vertical diameter and call its higher endpoint $D_2$. Then, repeat this process, constructing a circle $C_3$ centered at $D_2$ that passes through $A_2$ and $B_2$, drawing its horizontal and vertical diameters and constructing points $A_3$, $B_3$, and $D_3$ analogously, and so on until you construct $D_5$. What is the distance between $D_5$ and the center of $C_1$? 
 • **Needed insight.** There is a pattern to the construction, so that the distance between $C_1$ and $D_n$ is geometric in n, which allows you to avoid actually constructing most of the circles. |

## C  Experiment Details

### C.1  Experimental Setup

**Models.**  All experiments are conducted using the base model *Qwen2.5-7B-Instruct*, a strong instruction-tuned large language model. This model serves as the initialization for reinforcement learning (RL) fine-tuning.

**Datasets.** The training and evaluation problems for explorative, compositional, and transformational generalization are drawn from the curated problem families described in Appendix B. Unless otherwise specified, each training set consists of 1,000 problems. For compositional settings where training involves two problem families, we allocate 500 samples per family. To align with the proficiency level of *Qwen2.5-7B-Instruct*[6], the training problems are restricted to complexity levels 1–2. Evaluation is performed on:

- **In-distribution (ID)** problems: 100 test samples drawn from the same complexity range (1–2) as training, depending on the setup—whether explorative, compositional, or transformational.
- **Explorative** problems: 100 test samples from the same problem family within the explorative problems but with higher complexity (level 3).
- **Compositional and Transformational** problems: 20–50 test samples per setting. Although these problems do not have explicit complexity annotations, we adjust key parameters (like from small to large) to ensure the test set spans a range of complexity.

**Training Details.** We fine-tune models using the GRPO algorithm implemented in the Open-Instruct framework[7]. The key training parameters are as follows:

```
--beta 0.0
--num_unique_prompts_rollout 128
--num_samples_per_prompt_rollout 64
--kl_estimator kl3
--learning_rate 5e-7
--max_token_length 8192
--max_prompt_token_length 2048
--response_length 6336
--pack_length 8384
--apply_r1_style_format_reward True
--apply_verifiable_reward True
--non_stop_penalty True
--non_stop_penalty_value 0.0
--chat_template_name r1_simple_chat_postpend_think
--temperature 1.0
--masked_mean_axis 1
--total_episodes 20000000
--deepspeed_stage 2
--per_device_train_batch_size 1
--num_mini_batches 1
--num_learners_per_node 8 8
--num_epochs 1
--vllm_tensor_parallel_size 1
--vllm_num_engines 16
--lr_scheduler_type linear
--seed 3
--num_evals 200
```

**Evaluation Protocol.** Evaluation uses the same sampling strategy as training. Models are evaluated 200 times throughout training. To account for convergence fluctuations, we report the average performance over the last 5 evaluation checkpoints.

---

[6]Successful RL training requires the base model to achieve nonzero accuracy on the training problems.

[7]https://github.com/allenai/open-instruct

**Compute Resources.** Each RL training run uses 32 NVIDIA H100 GPUs (distributed across 4 nodes) and completes in approximately 12 hours.

## C.2    Prompt for Reasoning Trace Step Classification

To systematically analyze the types of reasoning exhibited in model-generated mathematical traces, we employed a structured prompt to guide the annotation of each sentence within the reasoning chain. This prompt instructs the LLM to classify each sentence into one of three categories—*conjecture*, *computation*, or *other*—with further verification for the correctness of computational steps.

The full prompt is as follows:

```
You are analyzing a sentence from a mathematical reasoning trace.
Please classify the following sentence into one of these categories:

1. "conjecture" - The sentence makes a hypothesis or conjecture about the final
answer. Typical examples include "Alternatively, maybe the matrix is singular.",
"Wait, let's check if the determinant is zero or not.", "Alternatively, maybe
the problem is from a source where the answer is 14."
2. "computation" - The sentence performs a mathematical computation or calculation.
3. "other" - The sentence is explanation, setup, conclusion, or another type of reasoning.

Original math problem: {original_question}
Correct answer: {correct_answer}
Sentence to classify: {sentence}

If you classify it as "computation", also verify if the computation is correct
by doing the calculation yourself.

Please respond in the following JSON format:
{
    "classification": "conjecture|computation|other",
    "reasoning": "Brief explanation of why you classified it this way. ",
    "computation_correct": true/false/null (only fill if classification is "computation")
}
```

This prompt enables fine-grained, reproducible labeling of reasoning steps for downstream analysis. In our experiments, we applied it to every step separated with ".\n" of the chain-of-thought traces.

# D  Additional Experiments

**Chain-of-Thought reasoning patterns analysis.** we observed several key patterns: **i) early solution discovery followed by excessive verification:** CoT traces in correct answers reveal that models often reach correct solutions relatively early in their responses but then spend substantial additional tokens on verification and double-checking. The yellow "overthinking" regions in Figure 9 show this post-solution elaboration, which remains consistent across most domains, though it can increase with task complexity (e.g., in algebra, up to 3k extra tokens spent on verification) even when the answer has already been found. Spending more tokens to verify an answer can be beneficial, but models must be cautious as excessive elaboration may introduces unnecessary steps, increases compute cost, and can destabilize otherwise correct reasoning. **ii) overthinking leads to spiral loops of errors:** we noticed that incorrect responses consistently consume more tokens than correct ones across all complexity levels. Response length initially increases with problem complexity, but then drops for some tasks at the highest levels and models abandon systematic reasoning when prob-

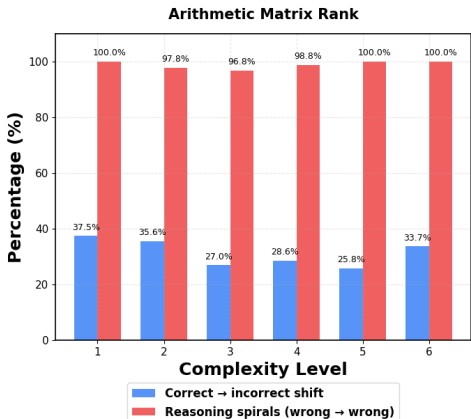

Figure 8: The percentage of incorrect responses exhibiting two distinct error patterns: correct → incorrect shift (blue bars) where models initially provided correct answers but changed to incorrect ones through overthinking, and reasoning spirals (red bars) where models remained in wrong → wrong reasoning chains throughout their response.

lems become intractable. To understand the types of reasoning failures models exhibit, we identified two dominant patterns (Figure 8). The first is the *correct → incorrect* shift, where models initially arrive at the correct answer but then second-guess themselves and revise toward an incorrect one (∼38% of incorrect responses at complexity 1, with similar trends across higher levels). The second is *reasoning spirals* (*wrong → wrong*), in which models never reach the correct answer and instead cycle through multiple flawed reasoning paths, making repeated errors without converging. This reveals that CoT with self-correction and backtracking, although significantly beneficial, is not sufficient to counter the snowballing of errors—transformers' autoregressive nature still compounds early mistakes, and CoT overthinking can paradoxically lead models to abandon the correct branch and answer, causing them to fall into spirals of errors.

**Is Lower Accuracy Simply Caused by Errors in Computation? Not Really, LLMs Exhibit Preference for Heuristics Over Direct Computation** Earlier we observed a steady decline in solution accuracy as the complexity level of our benchmarks rises. A plausible explanation is that harder problems require longer numeric derivations which amplifies the chance of arithmetic slips [33]. To disentangle cause from correlation, we zoom in on the *Matrix Rank* family, whose solution path (Gaussian elimination) is mostly deterministic and whose intermediate results can be easily verified. For every DeepSeek-R1's trajectory that produced an *incorrect* final answer, we segmented the CoT at each line break and asked `O4-mini` to label each segment as (*i*) a **conjecture**—a speculative statement about the final answer, (*ii*) a **computation**—an explicit algebraic or numeric operation, or (*iii*) **other**. When a segment was tagged as a computation, we further checked whether its arithmetic was correct. We provide the prompt details in Appendix §C.2.

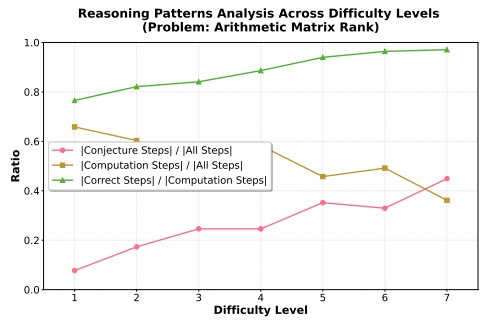

Figure 10: Reasoning trace analysis with distribution of two specific types of reasoning steps and correctness for the computation step, tested on *Matrix Rank* problem family. As problem difficulty increases, the model spends *less* of its CoT on explicit calculations (gold squares) and *more* on conjectural guesses (pink circles).

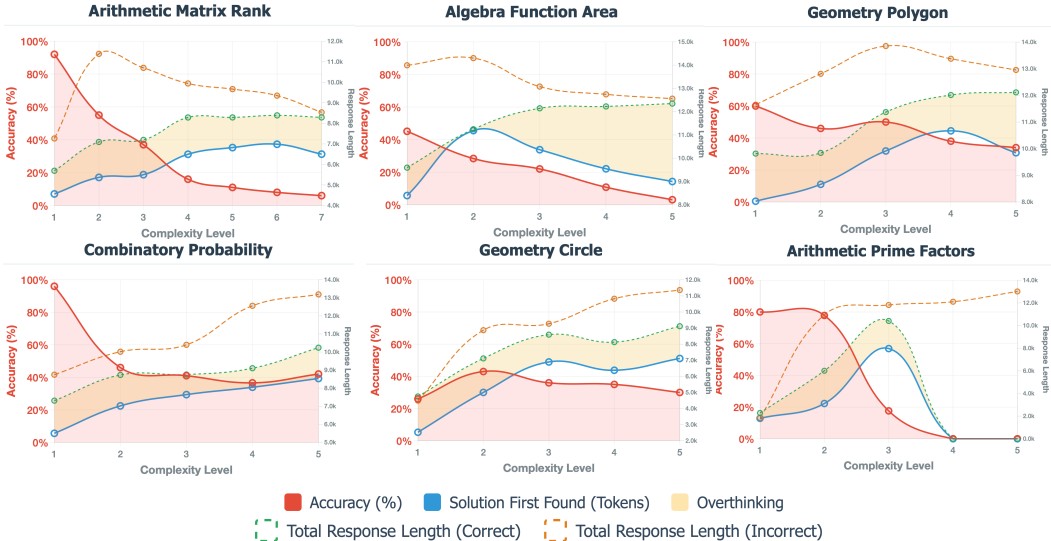

Figure 9: Performance and reasoning patterns across six mathematical task domains showing accuracy degradation and verification behavior as problem complexity increases. Models often reach the correct answer early in the response but continue generating unnecessary verification steps, as shown in the yellow overthinking regions. This behavior increases token usage and can destabilize otherwise correct outputs. Incorrect responses consistently consume more tokens than correct ones.

Figure 10 reveals three key trends: a) **Shrinking calculation budget.** The fraction of tokens devoted to actual computation drops from roughly 65 % at level 1 to below 40 % at level 7. b) **Growing reliance on guesswork.** Conjectural statements expand to fill the gap, indicating that the model increasingly tends to "jump to an answer" instead of working it out. c) **High per-step accuracy.** Paradoxically, when the model does compute, it does so *more* reliably at higher levels which suggests that arithmetic precision may not be the only bottleneck.

Collectively, these patterns show that the accuracy loss at higher complexity can be not only driven by cascading numerical mistakes, but also by the model's reluctance to invest reasoning budget in systematic calculation. Mitigating this issue may therefore require steering mechanisms that incentivize faithful computation rather than merely improving arithmetic skill.

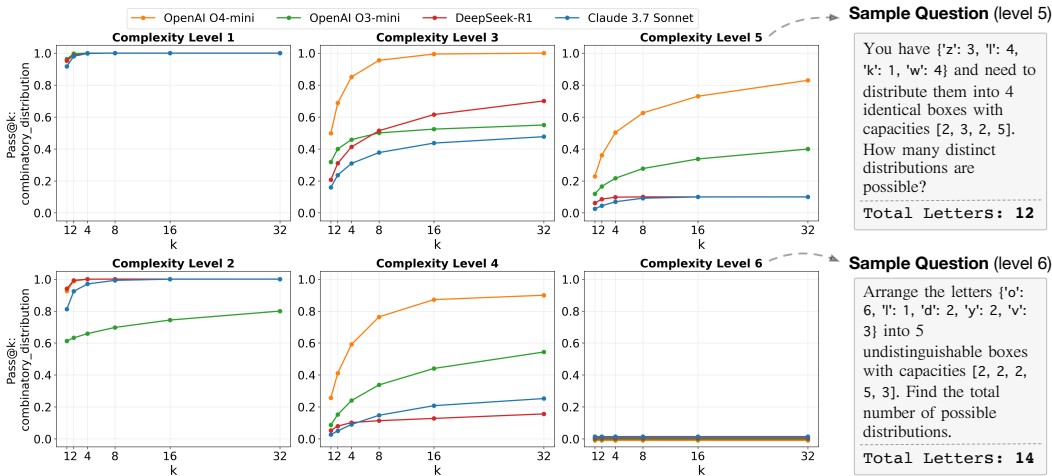

Figure 11: Pass@k performance of the advanced LLMs across complexity levels for geometry rotation problems.

**Can More Inference-Time Compute Solve Harder Problems? Helps at Moderate Complexity, but Gains Plateau at Higher Levels**    To investigate how inference-time compute contributes to

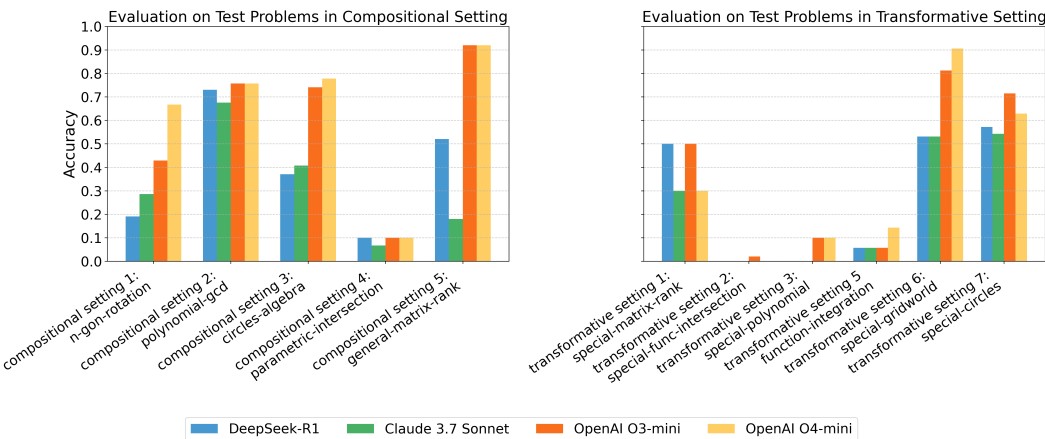

Figure 12: Performance comparison of state-of-the-art LLMs on mathematical reasoning tasks in compositional (left) and transformative (right) settings.

solving difficult math problems, we scale the number of candidates at inference time from 1 to 32 for the advanced LLMs and report *Pass@k* across six graded complexity levels. Figure 11 shows results for the "*letter distribution*" problems (we provide more problems in Figure 13 in Appendix §D). Results show that increasing the search space improves performance, gradually approaching 100% when the problem complexity is low. However, as the complexity increases, the benefit diminishes, and performance drops to zero at complexity level 6. Notably, this failure is not due to context length limitations—when solving this problem with dynamic programming, the state space remains within the LLM's context window. For example, the level 6 question example in Figure 11 only takes 36 unique states in using a DP solver. This abrupt failure underlines how a seemingly modest increase in combinatorial load can overwhelm current reasoning LLMs, highlighting that increasing the search space cannot necessarily mitigate the fundamental limits of transformers. While brute force helps, there must be smarter scaling approaches so that models learn the underlying algorithms and skills to solve math problems rather than simply relying on increased compute. Due to budget constraints, we limited testing to 64 attempts, but given the zero performance, we speculate that increasing beyond this point would not help.

**Frontier Models' Performance on the Test Problems in Compositional/Transformative Setting.**
We provide results in Figure 12. In the compositional setting, OpenAI models (particularly o4-mini and o3-mini) demonstrate superior performance on structured problems like matrix rank and polynomial operations, suggesting strong capabilities in combining fundamental mathematical concepts. Claude 3.7 Sonnet and DeepSeek-R1 show more moderate performance in this setting. In the transformative setting, all models struggle with special function intersections and certain polynomial problems. These results highlight both the progress made in LLMs' mathematical reasoning and the remaining challenges in developing models capable enough in different mathematical contexts.

**Ablation Study on Disentangling the Role of In-distribution Problem Family in Compositional RL Gain.** To better understand under which in-distribution problem family RL improves performance on compositional test problems, we conduct an ablation study (see Table 11 and Table 12) on the two compositional settings (Settings 2 and 5) that showed notable gains after RL fine-tuning according to Figure 7. In these settings, the model was originally trained jointly on two distinct problem families (`skill A` and `skill B`), and tested on composite tasks that require integrating both skills. Since not all settings benefited from RL, we hypothesize that the specific choice and compatibility of skill A and skill B may influence whether RL can effectively promote compositional generalization.

To test this hypothesis, we retrain the model in each setting while systematically altering the composition: replacing either `skill A` or `skill B` with a nearby alternative, or replacing both. Results show that the original `skill A + skill B` pairing consistently yields the highest post-RL improvement (+7.5 pp and +15 pp), indicating a strong synergy between the selected task pairs. Replacing just one component reduces gains to a modest +2–5 pp, while replacing both typically eliminates or reverses

improvement (-18 pp and -3 pp). These findings suggest that RL is most effective when it can build upon complementary skills already aligned in the joint training distribution—supporting the idea that compositional success depends not just on RL, but on the semantic coherence of the underlying task pair.

Table 11: Ablation study for **Compositional Setting 2** corresponding to Figure 7. All numbers are accuracies (0–1). Δ = After RL – Before RL.

| Training Composition (ID1 + ID2) | Before RL | After RL | Δ |
|---|---|---|---|
| *Original*: | | | |
| combinatory/prob_no_fixed + arithmetic/rank | 0.30 | 0.38 | **+0.08** |
| *Replace skill A*: | | | |
| combinatory/pattern_matching + arithmetic/rank | 0.30 | 0.35 | +0.05 |
| *Replace skill B*: | | | |
| combinatory/prob_no_fixed + arithmetic/GCD | 0.30 | 0.29 | -0.01 |
| *Replace both*: | | | |
| algebra/linear_equation + arithmetic/GCD | 0.30 | 0.12 | -0.18 |

Table 12: Ablation study for **Compositional Setting 5** corresponding to Figure 7.

| Training Composition (ID1 + ID2) | Before RL | After RL | Δ |
|---|---|---|---|
| *Original*: | | | |
| geometry/polygon_rotation + combinatory/pattern_matching | 0.05 | 0.20 | **+0.15** |
| *Replace ID1*: | | | |
| geometry/polygon_rotation + combinatory/distribution | 0.05 | 0.10 | +0.05 |
| *Replace ID2*: | | | |
| geometry/basic + combinatory/pattern_matching | 0.05 | 0.07 | +0.02 |
| *Replace both*: | | | |
| arithmetic/GCD + algebra/linear_equation | 0.05 | 0.02 | -0.03 |

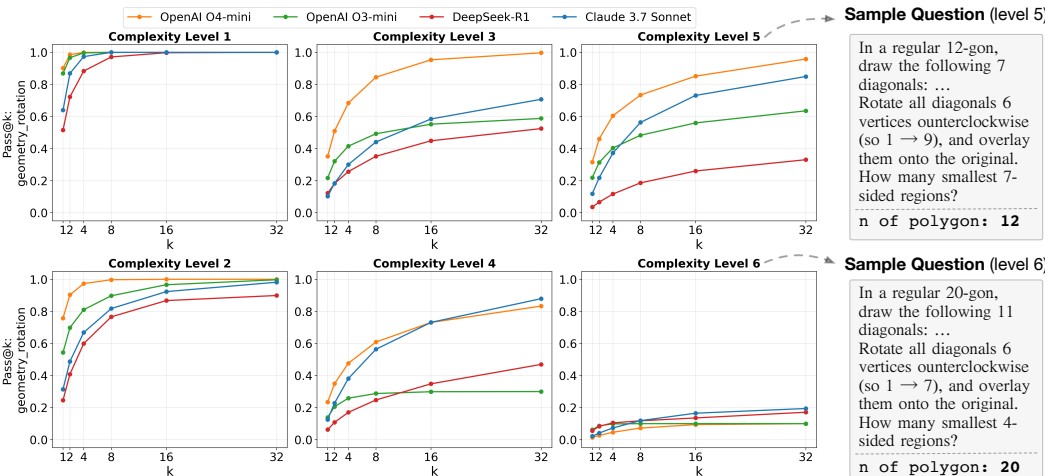

Figure 13: Pass@k performance of the advanced LLMs across complexity levels for geometry rotation problems.

**Supplementary Analysis on Qwen2.5-Math-7B.**  As shown in Figure 14, RL fine-tuning consistently improves performance on both in-distribution and explorative generalization tasks, with Qwen2.5-Math-7B achieving average gains of +51 percentage points on ID problems and +24 percentage points on OOD problems. Notably, the Math-7B model demonstrates particularly strong performance on Logic Zebralogic, reaching 85% ID accuracy and 82% OOD accuracy after RL training—indicating that the specialized mathematical training of the base model synergizes effectively with our RL approach. While Qwen2.5-7B-Instruct generally achieves slightly higher absolute performance (e.g., 95% vs 85% on Logic Zebralogic ID), both models exhibit similar improvement patterns, with consistently larger gains on ID tasks compared to OOD tasks. Interestingly, both

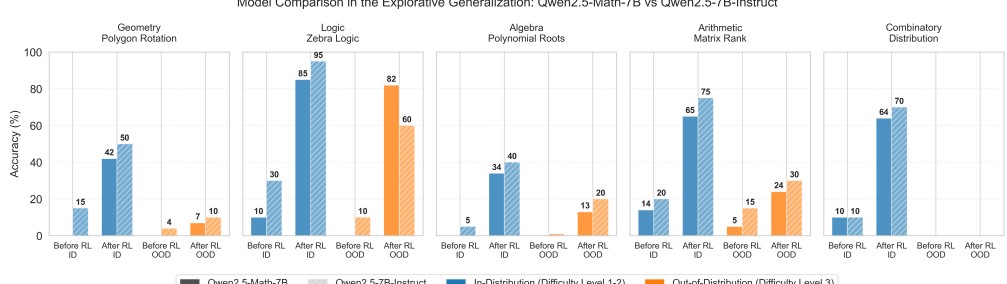

Figure 14: Comparison of RL fine-tuning effectiveness in the explorative generalization setting between *Qwen2.5-Math-7B* and *Qwen2.5-7B-Instruct*. Accuracy on in-distribution (ID) and out-of-distribution (OOD) mathematical reasoning tasks before and after RL fine-tuning. Solid bars: Math-7B; hatched bars: Instruct-7B.

models struggle with OOD generalization on the Combinatory Distribution task (0% OOD accuracy for both), suggesting this represents a particularly challenging generalization scenario that warrants further investigation. These results demonstrate that our RL fine-tuning methodology generalizes effectively across different Qwen2.5 variants, supporting the broader applicability of the approach for enhancing mathematical reasoning capabilities.

# E    Complexity Analysis

In this section, we present a complexity analysis for the COMBINATORY/DISTRIBUTION task, which is studied often in the main paper. Our goal is to demonstrate that this problem can be solved within the context-length limits of today's frontier large language models and Qwen-series models. Unlike other tasks, such as function intersection or geometry, where the number of tokens required is difficult to estimate, combinatory distribution problems allow for more precise tracking via simulation using a Python program. Our analysis proceeds in three steps: (i) we summarize the context window limits of current large-context models; (ii) we provide a representative level-6 problem and a compact dynamic programming (DP) solver; and (iii) we measure the solver's computational footprint and estimate the corresponding token usage.

## E.1    Context windows of frontier models

| Model | Maximum tokens | Reference |
|---|---|---|
| GPT-o3 Mini | 200 000 | OpenAI Docs |
| GPT-o4 Mini | 128 000 | Addepto Blog |
| Claude 3 Sonnet v3.7 | $\geq$200 000 | Anthropic Support |
| DeepSeek-R1 | 164 000 | OpenRouter Card |

Table 13: Context-length limits of the models considered in this work.

## E.2    Representative level-6 problem

> *Arrange the letters* $\{$o:6, l:1, d:2, y:2, v:3$\}$ *into five indistinguishable boxes with capacities* $[2, 2, 2, 5, 3]$. *How many distinct distributions exist?*

This family generalises classical balls-into-bins counting with (i) multisets of item types and (ii) capacity constraints. Difficulty level $k$ controls the total number of items and the size of the search space; level 6 is the hardest setting used in our experiments.

### E.3 DP solver and instrumentation

We employ a depth-first DP that memoises states of the form $(i, c)$, where $i$ indexes the current letter type and $c$ is the non-increasing vector of residual capacities. The core Python routine is shown below. Four counters track its execution:

- `dp_calls` – total invocations of the memoised routine;
- `distribution_calls` – number of distinct "distribute $t$ items into $c$" sub-problems generated;
- `backtrack_calls` – recursive steps inside the enumerator;
- `state_transitions` – edges explored in the DP graph.

```
def gen_distributions(total, rem_caps):
    global distribution_calls
    distribution_calls += 1
    # return all possible distributions of 'total' items into boxes with caps rem_caps
    # Generate recursively or via DP.
    # Use backtracking: assign to box 0 0..min(total,cap), then recuse.
    n = len(rem_caps)
    dist = []
    def backtrack(i, remaining, current):
        global backtrack_calls
        backtrack_calls += 1
        if i==n:
            if remaining==0:
                dist.append(tuple(current))
            return
        cap = rem_caps[i]
        # for each assign 0 to min(remaining,cap)
        for x in range(min(remaining,cap)+1):
            current.append(x)
            backtrack(i+1, remaining-x, current)
            current.pop()
    backtrack(0, total, [])
    return dist

@lru_cache(None)
def dp(i, rem_caps):
    global dp_calls, state_transitions
    dp_calls += 1
    if i == len(letter_counts):               # base case
        return 1
    total = letter_counts[i]
    count = 0
    for dist in gen_distributions(total, rem_caps):
        state_transitions += 1
        new_caps = tuple(sorted(rem_caps[j] - dist[j]
                                for j in range(len(rem_caps))))
        count += dp(i + 1, new_caps)
    return count
```

### E.4 Empirical resource usage

Running the solver on the level-6 instance yields the statistics in 14. The backtracking routine dominates runtime with 1059 calls. Conservatively assuming that *each* backtrack call translates to 20 generated/consumed tokens, the total token demand is

$$1059 \times 20 = 21\,180 \text{ tokens,}$$

well below even the smallest window in Table 13. Other problems in the same problem family with complexity level 6 exhibit similar footprints (average 1284 backtrack calls).

| Counter | Value | Explanation |
|---|---|---|
| `Unique DP states` | 36 | Distinct $(i, c)$ pairs memoised |
| `dp_calls` | 36 | Matches number of unique states |
| `distribution_calls` | 35 | Sub-problems created by the enumerator |
| `backtrack_calls` | 1059 | Leaf-level enumeration steps |
| `state_transitions` | 249 | Edges traversed in DP graph |

Table 14: Execution statistics for the level-6 exemplar.

## E.5 Footprint across difficulty levels (1–5)

We measured the average number of backtrack calls on the canonical instance for each lower difficulty.
Table 15 summarises these, along with the corresponding token estimates:

| Level | Avg. backtrack calls | Tokens@20/call | Estimated total tokens |
|---|---|---|---|
| 5 | 701.6 | $701.6 \times 20$ | 14 032 |
| 4 | 443.7 | $443.7 \times 20$ | 8 874 |
| 3 | 179.7 | $179.7 \times 20$ | 3 594 |
| 2 | 65.1 | $65.1 \times 20$ | 1 302 |
| 1 | 19.2 | $19.2 \times 20$ | 384 |

Table 15: Average backtracking calls and estimated token usage for levels 1–5.

Even at level 5—the hardest below level 6—the solver requires only $\approx$14 K tokens. All levels thus
comfortably fit within every model's context window, confirming the practicality of our experiments.

## E.6 Take-away

Even under pessimistic token-accounting assumptions, level-6 COMBINATORY DISTRIBUTION
problems demand fewer than 30 000 tokens of "reasoning budget". All four frontier models listed in
Table 13 therefore possess ample context to solve every instance we evaluate, validating the feasibility
of our experimental design.

