# OpenReview forum: "OMEGA: Can LLMs Reason Outside the Box in Math? Evaluating Exploratory, Compositional, and Transformative Generalization"
_NeurIPS.cc/2025/Datasets_and_Benchmarks_Track — NeurIPS 2025 Datasets and Benchmarks Track poster_

### Official Review · Reviewer_musm · 2025-06-21

**Rating:** 4
**Confidence:** 4

**Summary:**

The authors propose MathOOD dataset, with both training and test examples to evaluate LLMs' OOD math reasoning from three different angles.
They derive the problems from templated problem generators across six different domains.
They conduct experiments of both SFT and RL with Qwen-series models.

**Dataset Code Accessibility:**

Yes

**Ethical Considerations:**

No, there are no or only very minor ethics concerns

**Final Justification:**

All concerns are addressed throughout discussion with authors.

**Limitations Weaknesses:**

1. Lack of many important details in the main text, making the writing a little bit confusing:

+ The MathOOD is claimed to be constructed via templated problem generators, but the main body does not provide the source of the problems. Where do you collect the source problems?

+ In L99-L100, you claim that the difficulty level is comparable to AIME. How do you guarantee this? Do you have human experts to solve these problems, and find that the solving rate is similar to that of AIME?

+  The relationship between the templated problem generators in Table 2 and the three categories is unclear. It seems that these templates are only used to construct "Exploratory" problems.

+ What is the evaluation setting? For example, it is unclear how you evaluate LLMs on MathOOD and what proportion of problems are used to evaluate each category. What are the decoding parameters you used to evaluate LLMs? Are you using the thinking mode or not?

+ The statistics of the constructed MathOOD are unknown, e.g., how many problems are in each category? How many examples are for testing and how many for training? What is the average tokens of the problem?

2. In SFT and RL, you only use Qwen-serires models; more LLMs need to be incorporated to strengthen your conclusions.

3. About the category "transformative":  What if a problem has multiple different solutions (different strategies that lead to the same answer), how do we define "needed insight"?

4. The authors seem unfamiliar with related works:

+ Some important related works are not covered:
  - In L41 - 42: "GSM-Symbolic [25 ], GSM-PLUS [ 22], and GSM-Infinite [ 39] focus on narrow domains making the diversity of reasoning problems limited." Some works functionize the MATH dataset, which is more diverse (covers 7 subjects in math) (e.g., [1]).

  - For works like GSM-PLUS, the authors do not cover similar works in the related works. To name a few, GSM-IC [2], VarBench [3], E-GSM [4].

 - L38-41: "blend a large number of math questions in different topics and difficulty levels, making it hard to isolate specific reasoning skills behind a model’s success or failure." Actually, Omni-Math and other works (e.g, OlympicArena [5]) have fine-grained metadata (e.g., difficulty levels or cognitive capabilities) to isolate different skills behind a model's success or failure.

- In L37 - 46, Table 1 and L259 - 269, the authors mix training sets (e.g., DeepMath-103K, Numina-Math) and test sets (e.g., Omni-Math, Math500). It is quite unprofessional.

5. Others:

+ In L1, and L25: Large-scale Language Models -> Large Language Models. I suggest the authors double-check the polished text returned by LLMs.

+ In Table2, the column "$\delta (\theta)$" is not described in the captioned. It is discussed in Section 2.2, but Table 2 is not cross-referenced in Section 2.2.

+ I do not know why Section 2.2 is named "Training and Evaluation Setup for Generalization". In fact, no details of training and evaluation setup are provided except the complexity measure.



[1] Srivastava, S., PV, A., Menon, S., Sukumar, A., Philipose, A., Prince, S., & Thomas, S. (2024). Functional benchmarks for robust evaluation of reasoning performance, and the reasoning gap. arXiv preprint arXiv:2402.19450.

[2] Shi, F., Chen, X., Misra, K., Scales, N., Dohan, D., Chi, E. H., ... & Zhou, D. (2023, July). Large language models can be easily distracted by irrelevant context. In International Conference on Machine Learning (pp. 31210-31227). PMLR.

[3] Qian, K., Wan, S., Tang, C., Wang, Y., Zhang, X., Chen, M., & Yu, Z. (2024). VarBench: Robust language model benchmarking through dynamic variable perturbation. arXiv preprint arXiv:2406.17681.

[4] Xu, X., Xiao, T., Chao, Z., Huang, Z., Yang, C., & Wang, Y. (2024). Can LLMs Solve longer Math Word Problems Better?. arXiv preprint arXiv:2405.14804.

[5] Huang, Z., Wang, Z., Xia, S., Li, X., Zou, H., Xu, R., ... & Liu, P. (2024). Olympicarena: Benchmarking multi-discipline cognitive reasoning for superintelligent ai. Advances in Neural Information Processing Systems, 37, 19209-19253.

**Strengths Contributions:**

1. To evaluate the math reasoning abilities of LLMs in a controlled manner is interesting and meaningful.

2. The proposed MathOOD also has training examples, which are quite different from the current math benchmarks.

3. The experiments incorporate both SFT and RL, in some sense is comprehensive.

---

> ### Author Rebuttal · Authors · 2025-07-31
>
> **Opening Remark**: We thank the reviewer for the constructive and valuable feedback.
> We are honored that the reviewer finds our approach to evaluating mathematical reasoning in LLMs interesting and meaningful. We are equally pleased that our MathOOD dataset, with its distinctly different training examples compared to existing benchmarks, was positively recognized. Additionally, we appreciate the reviewer’s acknowledgment of the comprehensiveness of our experiments.
> We have carefully addressed the reviewer’s comments and concerns below.
>
> > **Source of Problem Generators**
>
> The majority of problem templates were curated by three co-authors with substantial experience in mathematics competitions, including AIME. In addition, a subset of logic and puzzle problems was adapted from established sources, such as zebra-logic puzzles. While some existing olympiad problems inspired our work, many required substantial re-derivation to ensure answers could be efficiently synthesized by programs and verified. We therefore derive those problems that can be synthesized. Notably, generating valid geometry problems in our paper required non-trivial development of a custom library supporting tree-based exploration of geometric configurations. More details on curation and generation procedures will be added to the main text.
>
> > **Justification of AIME-level Difficulty**
>
> One of our goals was to create a synthesized benchmark at a higher difficulty level than GSM-family datasets, which predominantly feature primary school-level word problems. By claiming AIME-level difficulty, we refer to high-school math competition standards. As empirical support, the DeepSeek-R1 model achieves 71% accuracy on our level-1 problems (see Figure 3), closely matching the official AIME 2024 average (78%), suggesting comparable challenge and selectivity.
>
> > **Relationship Between Problem Templates and Generalization Axes**
>
> The problem templates are used across all training stages for the three generalization axes, depending on the specific settings. In the test time, it is also used for the exploratory generalization setting—test samples are drawn from higher difficulty levels of the same template family. For compositional and transformative generalization, test questions are curated separately by combining or altering base templates to ensure the required novel integration or strategy shift.
>
>
> > **Evaluation Settings**
>
>
> Training and evaluation protocols are detailed in Section 3.2 and Appendix B; a summary is provided here for convenience:
>
> **Datasets**. Each training set consists of 1,000 problems. For compositional settings where training involves two problem families (ID1 and ID2), we allocate 500 samples per family. To match the proficiency level of Qwen2.5-7B-Instruct, training problems are restricted to difficulty levels 1–2. Evaluation is performed on:
> - In-distribution (ID) problems: 100 test samples drawn from the same difficulty range (1–2) as training.
> - Explorative problems: 100 test samples from the same problem family but with higher difficulty (level 3).
> - Compositional and Transformational problems: 20–50 test samples per setting.
>
> **Decoding paramter**. The decoding parameter is setting to temperature 1.0 and other default setting in `vllm` framework.
>
> **Thinking mode usage**. We provide instruction in prompt that the thinking procedure should be wrapped in `<think></think>`, there is a formatting error used in RL training to instruct base model to follow this standard. More train/eval details are in the Appendix B.1 and https://github.com/allenai/open-instruct.
>
> > **Dataset Statistics**
>
> a) **Number of Problems Per Category**: Our generator allows for unlimited synthesis; for each experiment, sample counts are as above.
>
>
> b) **Token Statistics**: A detailed statistics are provided below:
>
> | Domain         | Mean Tokens | Median | Std Dev | Min | Max |
> |----------------|-------------|--------|---------|-----|-----|
> | Logic          | 153.4       | 114.0  | 94.4    | 87  | 579 |
> | Geometry       | 143.5       | 137.0  | 34.1    | 73  | 333 |
> | Combinatorics  | 99.3        | 112.0  | 23.5    | 52  | 122 |
> | Algebra        | 97.5        | 92.0   | 31.4    | 38  | 244 |
> | Arithmetic     | 78.7        | 77.0   | 40.1    | 27  | 259 |
> | Number Theory  | 76.9        | 85.0   | 11.9    | 47  | 88  |
>
>
> > **Additional Experiments on Non-Qwen Model Series**
>
> Thank you for your insightful suggestion. To address this concern, we extended our experiments to two additional LLM series—Gemma3 (Google) and Nemotron (Nvidia)—and evaluated their performance in the explorative, compositional, and transformative generalization settings (see the tables below, which correspond to Figure 4 in the paper).
>
> Our findings show that the overall trends observed in the Qwen-series models are robust across architectures: RL training predominantly enhances explorative generalization, while its effect on compositional and transformative reasoning remains limited. The consistency of these results strengthens the main conclusions of our work.
>
>
> #### **Gemma-3-4b-it**
> ##### Explorative Generalization
> |  | Algebra | Arithmetic | Combinatory | Geometry | Logic | Number Theory
> -|---------|-------|-------------|----------|-------|----------------
> Base   Model      |    6    |      10     |     8      |     5    |   9   |       4
> RL Improve   |  +13    |    +22     |    +19      |   +3     | +50   |      +10 |
>
> ##### Compositional Generalization
>
> |  | Comp. Setting 1 | Comp. Setting 2 | Comp. Setting 3 | Comp. Setting 4 | Comp. Setting 5
> --------------------------|-----------------|-----------------|-----------------|-----------------|-----------------
> Base Model          |       10        |        6        |    25        |        4        |       32
> RL Improve     |        0        |      +5        |        0        |       0        |       +10
>
> ##### Transformative Generalization
>
> | | Trans. Setting 1 | Trans. Setting 2 |	Trans. Setting 3 | Trans. Setting 4 |
> |-------------------------|----------------|------------------|------|----------|
> | Base Model |	45 | 0 | 2	|  0 |
> | RL Improvement |	+ 6 | 0 | 0  |  0 |
>
>
> #### **OpenReasoning-Nemotron-7B**
> ##### Explorative Generalization
> |                     | Algebra | Arithmetic | Combinatory | Geometry | Logic | Number Theory|
> ---------------------|---------|------------|-------------|----------|-------|------|
> Base   Model      |    26    |      37     |     18      |     12    |   45   |       20  |
> RL Improve   |  +13    |    +22     |    +10      |   +5     | +17   |      +12 |
>
> ##### Compositional Generalization
> |  | Comp. Setting 1 | Comp. Setting 2 | Comp. Setting 3 | Comp. Setting 4 | Comp. Setting 5
> ---|-----------------|-----------------|-----------------|---|--
> Base Model          |       10        |        12        |    25        | 12        |       36
> RL Improve     |        0        |      +6        |        0        |      +2        |       +8
>
> ##### Transformative Generalization
>
> | | Trans. Setting 1 | Trans. Setting 2 |	Trans. Setting 3 | Trans. Setting 4 |
> |-------|----------------|---|--|--------|
> | Base Model |	75 | 0 | 0	|  0 |
> | RL Change |	-20 | 0 | 0  |  0 |
>
>
> > **“Needed Insight” in Transformative Problems with Multiple Solution Strategies**
>
> Thank you for raising this important point. To address this, our approach focuses on deriving problems entailing the necessity of departing from familiar tactics. As discussed in Lines 179–181, `our test instances are specifically designed such that familiar or conventional strategies either outright fail or become intractably cumbersome.`
> In other words, although a problem might allow for different solution paths, each path still requires a shift away from standard reasoning. This is further illustrated in Table 3, where we show that the traditional approaches do not yield efficient or feasible solutions for these instances.
>
> > **Extended discussion on related works**
>
> We thank the reviewer for highlighting these important works. We will incorporate and discuss them more thoroughly in the revised version. Several recent benchmarks—such as MATH, GSM-IC, VarBench, and E-GSM—have introduced valuable perturbations or extensions to the MATH and GSM problem sets. While these datasets have helped drive progress, many state-of-the-art LLMs are now reaching near-saturation on them. In contrast, MathOOD employs programmatic generators to systematically scale up problem difficulty, presenting challenges even for top-performing models (see Figure 3).
>
> A key distinction of MathOOD lies in its explicit separation of generalization axes, enabling controlled train/test splits along exploratory, compositional, and transformative dimensions. This fine-grained experimental design provides a degree of control that is not achievable with the more coarse-grained tagging present in benchmarks like OlympicArena or Omni-Math. We will emphasize this unique aspect in our revision.
>
> Regarding the reviewer's comment on the mixing of training and test set discussions: Our benchmark explicitly defines both training and test splits, so our discussion references both in context. Nonetheless, we appreciate the suggestion and will clarify the purpose and organization of each dataset in our revision by tagging their main intended use (e.g., train, test, evaluation).
>
> > **Other Writing Glitches**
>
> We thank the reviewer for their careful reading. The following revisions have been made in response:
>
> - Changed all instances of "Large-scale Language Models" to "Large Language Models."
> - Added an explicit description of the column $\delta$ to the caption of Table 2 and ensured Table 2 is cross-referenced in Section 2.2.
> - Updated the title of Section 2.2 to "Overview of the Problem Generation and Generalization Setup," and clarified the content accordingly.
>
> We appreciate the reviewer’s attention to these details and have carefully addressed these points in the updated manuscript.

---

> > ### Comment · Reviewer_musm · 2025-08-02
> >
> > Thank you for your detailed reply and added experiments.
> >
> > I still have one question regarding the decoding parameters. Your rebuttal said, "The decoding parameter is setting to temperature 1.0 and other default setting in vllm framework." From the caption of Figure 3, you evaluated 100 samples and reported pass@1 for a single run. My concern is that for those reasoning models, pass@1 evaluated on only 100 samples with one single run might be very unstable. For evaluating reasoning models on AIME (30 examples), a common practice is to get 32/64 generations to report pass@1. For 100 examples, at least 4 generations will be needed.
> >
> > If the cost of calling APIs of these models is too high, I recommend that you use an open-source reasoning model to check the variation of "pass@1 for only one run" (i.e., run several times and compare pass@1 across different runs).

---

> > > ### Author Response · Authors · 2025-08-04
> > > **Response**
> > >
> > > Great suggestions! We are glad our previous response was helpful, and we appreciate your emphasis on evaluating the stability of this metric.
> > >
> > > To address your concern, we have conducted an updated analysis for Figure 3 by running 5 generations over 100 samples. We appreciate your recommendation and will include this additional analysis in the revised version of our paper.
> > >
> > > | **Problem**                          | **Model**         | **D1** | **D2** | **D3** | **D4** | **D5** |
> > > |--------------------------------------|-------------------|--------|--------|--------|--------|--------|
> > > | **algebra\_func\_area**              | DeepSeek-R1       |     40 |     27 |     18 |     10 |      6 |
> > > |                                      | Claude 3.7 Sonnet |     34 |     18 |     19 |     11 |      6 |
> > > |                                      | OpenAI O3-mini    |     59 |     38 |     28 |     19 |     10 |
> > > |                                      | OpenAI O4-mini    |     47 |     45 |     38 |     25 |     19 |
> > > | **algebra\_func\_intersection**      | DeepSeek-R1       |     29 |     17 |     14 |     10 |     10 |
> > > |                                      | Claude 3.7 Sonnet |     42 |     28 |     27 |     16 |      8 |
> > > |                                      | OpenAI O3-mini    |     65 |     43 |     41 |     37 |     17 |
> > > |                                      | OpenAI O4-mini    |     71 |     53 |     45 |     31 |     23 |
> > > | **combinatory\_distribution**        | DeepSeek-R1       |     83 |     78 |     71 |     11 |      3 |
> > > |                                      | Claude 3.7 Sonnet |     76 |     67 |     66 |      5 |      0 |
> > > |                                      | OpenAI O3-mini    |     78 |     80 |     74 |      7 |      4 |
> > > |                                      | OpenAI O4-mini    |     77 |     74 |     73 |     41 |     29 |
> > > | **arithmetic\_list\_prime\_factors** | DeepSeek-R1       |     98 |     76 |     19 |      2 |      1 |
> > > |                                      | Claude 3.7 Sonnet |     99 |     43 |      7 |      1 |      0 |
> > > |                                      | OpenAI O3-mini    |    100 |     97 |     73 |     38 |      1 |
> > > |                                      | OpenAI O4-mini    |     98 |     97 |     66 |      8 |      0 |
> > > | **arithmetic\_matrix\_rank**         | DeepSeek-R1       |     89 |     54 |     38 |     16 |     16 |
> > > |                                      | Claude 3.7 Sonnet |     73 |     36 |     33 |     19 |     17 |
> > > |                                      | OpenAI O3-mini    |    100 |     70 |     62 |     50 |     27 |
> > > |                                      | OpenAI O4-mini    |    100 |     77 |     65 |     34 |     16 |
> > > | **numbertheory\_lte\_qr**            | DeepSeek-R1       |     90 |     82 |     66 |     50 |     18 |
> > > |                                      | Claude 3.7 Sonnet |     85 |     97 |     65 |     51 |     31 |
> > > |                                      | OpenAI O3-mini    |     65 |     73 |     58 |     52 |     12 |
> > > |                                      | OpenAI O4-mini    |     67 |     52 |     53 |     50 |     19 |
> > > | **geometry\_circle**                 | DeepSeek-R1       |     71 |     38 |     18 |      7 |      0 |
> > > |                                      | Claude 3.7 Sonnet |     52 |     41 |     18 |     21 |     10 |
> > > |                                      | OpenAI O3-mini    |     80 |     68 |     57 |     19 |      8 |
> > > |                                      | OpenAI O4-mini    |     97 |     80 |     71 |     58 |     52 |
> > > | **geometry\_polygon**                | DeepSeek-R1       |     38 |      0 |      0 |      0 |      0 |
> > > |                                      | Claude 3.7 Sonnet |     20 |      11 |      10 |     8 |     5 |
> > > |                                      | OpenAI O3-mini    |     38 |     21 |     12 |      0 |      0 |
> > > |                                      | OpenAI O4-mini    |     59 |     55 |     27 |     11 |     10 |
> > > | **gridworld\_rookmove**              | DeepSeek-R1       |     79 |     18 |      0 |      0 |      0 |
> > > |                                      | Claude 3.7 Sonnet |     80 |     40 |     27 |     21 |      0 |
> > > |                                      | OpenAI O3-mini    |    100 |     97 |     60 |     39 |     25 |
> > > |                                      | OpenAI O4-mini    |     90 |     93 |     51 |     41 |     29 |
> > > | **gridworld\_blocked**               | DeepSeek-R1       |     90 |      7 |      2 |      0 |      0 |
> > > |                                      | Claude 3.7 Sonnet |     91 |      8 |      2 |      1 |      0 |
> > > |                                      | OpenAI O3-mini    |     97 |     98 |     69 |     9 |      0 |
> > > |                                      | OpenAI O4-mini    |    100 |     97 |     71 |      0 |      0 |

---

> > > > ### Comment · Reviewer_musm · 2025-08-04
> > > >
> > > > Thank you for your added experiments!
> > > >
> > > > Will such variability influence the conclusion of SFT and RL? It appears that different runs exhibit quite distinct behaviors. Maybe also updating Figure 4 results of pass@1 with 4 generations will be a good choice.
> > > >
> > > > Hope the authors will update this at a later stage if possible.
> > > >
> > > > I will update my score to 4 soon.

---

> > ### Author Response · Authors · 2025-08-04
> > **Response**
> >
> > Thank you for your thoughtful feedback!
> >
> > The evaluation results in Figure 4 are already reported as the average performance over 5 runs. To improve clarity, we will move these details from Appendix B to the caption of Figure 4.
> >
> > To better illustrate the stability of the results, we have also included fluctuation ranges, indicating the standard deviation across the 5 runs. The detailed results with standard deviations are presented below:
> >
> > |                      | Algebra | Arithmetic | Combinatory | Geometry | Logic | Numbertheory |
> > |----------------------|---------|------------|-------------|----------|-------|---------------|
> > | ID Improvement       | +27 (±3)    | +37 (±4)   | +60   (±2)    | +18   (±4)  | +52 (±3)  | +33  (±4)      |
> > | OOD Improvement      | +16 (±1) | +24  (±6)   | +15  (±2)    | +5  (±0)   | +39  (±3) | +14     (±3)      |
> >
> > |                         | Comp. Setting 1 | Comp. Setting 2 | Comp. Setting 3 | Comp. Setting 4 | Comp. Setting 5 |
> > |-------------------------|----------------|------------------|------------------|------------------|------------------|
> > | ID1 Improvement         | +28    (±3)     | +16   (±2)       | +54    (±3)        | +54   (±2)         | +56  (±3)         |
> > | ID2 Improvement         | +32     (±2)     | +69    (±4)       | +49   (±2)        | +48  (±4)        | +56  (±3)         |
> > | OOD Improvement         | 0   (±0)        | +15      (±5)        | 0    (±0)          | +6    (±1)           | +8  (±0)          |
> >
> > |                         | Trans. Setting 1 | Trans. Setting 2 |	Trans. Setting 3 | Trans. Setting 4 |
> > |-------------------------|----------------|------------------|------------------|------------------|
> > | RL Improve (ID) |	+56	(±3) | +39 (±2)| +36 (±4)	| +58 (±3) |
> > | RL Change (OOD) |	-30 (±11) | 0 (±0)  | 0 (±0) |  0 (±0)
> >
> > In a nutshell, our findings show that the overall trends observed hold: predominantly enhancing explorative generalization, while its effect on compositional and transformative reasoning remains limited.

---

> > > ### Comment · Reviewer_musm · 2025-08-05
> > >
> > > Thank you! My concerns are fully addressed.

---

### Official Review · Reviewer_fv3V · 2025-06-28

**Rating:** 4
**Confidence:** 2

**Summary:**

This paper introduces MathOOD, a novel benchmark designed to systematically evaluate the OOD generalization capabilities of LLMs in the domain of mathematical reasoning.

**Dataset Code Accessibility:**

Yes

**Ethical Considerations:**

No, there are no or only very minor ethics concerns

**Final Justification:**

I appreciate the authors' detailed response. I think my score is appropriate, and I will maintain my positive attitude towards this submission.

**Limitations Weaknesses:**

1.  The definition of problem difficulty in MathOOD is primarily based on human-perceived computational complexity, such as the magnitude of numbers or the size of matrices. The paper lacks a clear justification for this choice of difficulty metric. It is possible that some of these settings merely lengthen the model's inference process without substantially increasing the intrinsic reasoning challenge.
2.  The six problem categories appear to be highly suitable for code-based solvers. In a code-execution paradigm, simply adjusting numerical ranges may not effectively increase problem difficulty. If the task only involves more repetitive computational steps, it may not be an ideal approach for evaluating a model's mathematical reasoning capabilities.
3.  The empirical evaluation is limited to a few state-of-the-art (SOTA) large models. An analysis including smaller-scale models would be a valuable addition. Such a comparison could help determine whether certain problem types can be solved effectively by more compact models, thus clarifying the role of model scale.
4. The conclusions about the limitations of RL fine-tuning are based on experiments with a single model architecture (Qwen2.5-7B-Instruct). While insightful, this finding's generalizability to other model families, larger-scale models, or alternative training paradigms like naive SFT remains an open question. A larger model, for instance, might exhibit different emergent properties in compositional tasks.

**Strengths Contributions:**

1. The benchmark is constructed using 40 programmatically generated problem templates across six mathematical domains, enabling precise control over problem difficulty and structure.
2. By operationalizing cognitive science concepts of creativity into a concrete evaluation methodology, the authors move beyond standard benchmarks that often conflate various reasoning skills. The programmatic generation of problems from templates is a significant methodological advantage, affording a level of control and scalability that is absent in datasets scraped from static sources. This allows for clean, causal-style analysis of model capabilities, as demonstrated by the targeted train-test splits for each generalization axis.
3. The experimental design is particularly insightful; the RL fine-tuning experiment effectively isolates the effect of targeted training, leading to the compelling and well-supported conclusion that current training paradigms can enhance and extend existing skills but fail to induce the flexible skill integration or creative leaps characteristic of higher-level reasoning.
4.  The template-based generation of the evaluation data effectively mitigates the risk of data contamination, thereby ensuring the reliability of the benchmark.
5.  The experimental details are clearly documented in the appendix, which enhances the reproducibility of the results.

---

> ### Author Rebuttal · Authors · 2025-07-31
>
> **Opening Remark**: We thank the reviewer for the constructive and valuable feedback. We are honored that the reviewer recognizes the methodological rigor and precision achieved through our benchmark, constructed using programmatically generated problem templates spanning six mathematical domains. We are equally pleased that the reviewer appreciates our novel approach, operationalizing cognitive science concepts of creativity into a concrete evaluation methodology. Additionally, we value the acknowledgment of our insightful experimental design, which effectively isolates the impacts of targeted RL fine-tuning, as well as our efforts to ensure benchmark reliability and reproducibility through template-based data generation and comprehensive documentation. We have addressed the reviewer’s comments and concerns below.
>
>
> > **Clarification on Problem Difficulty and Intrinsic Reasoning Complexity**
>
> Great question! Our choice of difficulty levels aligns with the natural, intuitive complexity inherent in each problem family. For instance, in matrix computation tasks, increasing the dimensions naturally elevates computational complexity. Similarly, polygon-related problems become inherently more intricate as the number of vertices grows.
>
> As illustrated in Figure 3, the performance of four top-tier LLMs declines as problem complexity (defined in this intuitive manner) increases. This observed correlation underscores that our difficulty metric indeed captures genuine increases in intrinsic reasoning challenges, aligning well with human-perceived complexity.
>
>
> > **Suitability of Problem Categories for Code-Execution Paradigms**
>
> This is a fair concern, and we address it from several important angles:
>
> 1. **Fundamental Differences in Reasoning Approaches:** Mathematical and machine-computational solutions differ fundamentally. For example, combinatorial problems in mathematics typically involve strategic case analysis combined with permutation or combination formulas. In contrast, code-based solutions often default to brute-force enumeration. Similarly, finding function intersections mathematically involves derivative analysis and symbolic methods, whereas computational solutions resort to numerical approximations through extensive sampling.
> 2. **Limitations of Code-Execution Paradigm:** Many problem categories in this paper, such as geometry and logic puzzles, cannot be solved effectively or straightforwardly using code-execution alone. Examples include counting rectangles formed within an n-gon containing specific diagonals or solving logical puzzles involving complex relational constraints (e.g., determining house arrangements based on multiple interrelated clues).
> 3. **Evaluating "Brute-Force" Capabilities is One of the Purpose:** One explicit objective of our benchmark is precisely to assess the model's brute-force reasoning capabilities, particularly within the arithmetic/algebra category. Investigating how and why LLMs fail as complexity scales—even if the problem theoretically suits brute-force solutions—is nontrivial and insightful. The "exploratory generalization" dimension specifically targets this phenomenon, examining whether LLMs truly internalize algorithmic reasoning or merely replicate superficial computational steps.
>
> > **Evaluation of Smaller-Scale Models**
>
> In addition to state-of-the-art (SOTA) large models, we have included evaluations of smaller-scale (7B) models in Figures 4 (Qwen2.5-7B-Instruct) and 7 (Qwen2.5-Math-7B). While the primary purpose of these figures was to illustrate performance changes after reinforcement learning (RL), the solid bars represent the base-model performance. These results indicate generally inferior performance compared to commercial large-scale models, emphasizing the role model scale plays in handling complex mathematical reasoning tasks.
>
>
> > **Impact of Model Scale on RL Training Outcomes**
>
> To address the reviewer's suggestion regarding generalizability across model scales, we conducted additional experiments using a larger-scale model, Qwen2.5-32B-Instruct. Given computational constraints (each training scenario with GRPO requiring 72 H100 GPUs for two days), we present a focused analysis (on three settings) below:
>
> **Exploratory Generalization Test:** (Problem Family: arithmetic/matrix-Rank)
> | |  Qwen2.5-7B-Instruct |  Qwen2.5-32B-Instruct|
> |--|--|---------|
> | Base model performance (ID)      | 20 | 25 |
> | Improvement after RL (ID)      | +52   | +65 |
> | Base model performance (OOD)      | 15 | 18 |
> | Improvement after RL (OOD)     | +18   | +37 |
>
> **Compositional Generalization Test:** (Setting 5: combinatorics/prob-no-fix + arithmetic/matrix-rank)
> | |  Qwen2.5-7B-Instruct |  Qwen2.5-32B-Instruct|
> |--|--|---------|
> | Base model performance (ID1)      | 9 | 15 |
> | Improvement after RL (ID1)     | +56 | +70 |
> | Base model performance (ID2)       | 19 | 24 |
> | Improvement after RL (ID2)    | +56 | + 63 |
> | Base model performance (OOD)      | 30 | 32|
> | Improvement after RL (OOD)       | +8 | +10 |
>
> **Transformative Generalization test:** (Setting 4: combinatorics/prob-no-fix)
> | |  Qwen2.5-7B-Instruct |  Qwen2.5-32B-Instruct|
> |--|--|---------|
> | Base model performance (ID)      | 11 | 15 |
> | Improvement after RL (ID)      | +58 | +72 |
> | Base model performance (OOD)     | 0 | 0 |
> | Improvement after RL (OOD)       | +0 | +0 |
>
> These findings indicate a clear trend: the 32B model exhibits stronger baseline performance and larger gains in exploratory generalization after RL fine-tuning. However, improvements in compositional and transformative generalization remain limited, consistent with our original observations using the 7B model. This suggests that while model scale can amplify certain generalization capabilities, fundamental challenges persist in tasks requiring compositional and transformative reasoning.
>
>
> > **Additional Experiments on Non-Qwen Model Series**
>
> Thank you for your insightful suggestion. To address this concern, we extended our experiments to two additional LLM series—Gemma3 (Google) and Nemotron (Nvidia)—and evaluated their performance in the explorative, compositional, and transformative generalization settings (see the tables below, which correspond to Figure 4 in the paper).
>
> Our findings show that the overall trends observed in the Qwen-series models are robust across architectures: RL training predominantly enhances explorative generalization, while its effect on compositional and transformative reasoning remains limited. The consistency of these results strengthens the main conclusions of our work.
>
>
> #### **Gemma-3-4b-it**
> ##### Explorative Generalization
> |  | Algebra | Arithmetic | Combinatory | Geometry | Logic | Number Theory
> -|---------|-------|-------------|----------|-------|----------------
> Base   Model      |    6    |      10     |     8      |     5    |   9   |       4
> RL Improve   |  +13    |    +22     |    +19      |   +3     | +50   |      +10 |
>
> ##### Compositional Generalization
>
> |  | Comp. Setting 1 | Comp. Setting 2 | Comp. Setting 3 | Comp. Setting 4 | Comp. Setting 5
> --------------------------|-----------------|-----------------|-----------------|-----------------|-----------------
> Base Model          |       10        |        6        |    25        |        4        |       32
> RL Improve     |        0        |      +5        |        0        |       0        |       +10
>
> ##### Transformative Generalization
>
> | | Trans. Setting 1 | Trans. Setting 2 |	Trans. Setting 3 | Trans. Setting 4 |
> |-------------------------|----------------|------------------|------|----------|
> | Base Model |	45 | 0 | 2	|  0 |
> | RL Improvement |	+ 6 | 0 | 0  |  0 |
>
>
> #### **OpenReasoning-Nemotron-7B**
> ##### Explorative Generalization
> |                     | Algebra | Arithmetic | Combinatory | Geometry | Logic | Number Theory|
> ---------------------|---------|------------|-------------|----------|-------|------|
> Base   Model      |    26    |      37     |     18      |     12    |   45   |       20  |
> RL Improve   |  +13    |    +22     |    +10      |   +5     | +17   |      +12 |
>
> ##### Compositional Generalization
> |  | Comp. Setting 1 | Comp. Setting 2 | Comp. Setting 3 | Comp. Setting 4 | Comp. Setting 5
> ---|-----------------|-----------------|-----------------|---|--
> Base Model          |       10        |        12        |    25        | 12        |       36
> RL Improve     |        0        |      +6        |        0        |      +2        |       +8
>
> ##### Transformative Generalization
>
> | | Trans. Setting 1 | Trans. Setting 2 |	Trans. Setting 3 | Trans. Setting 4 |
> |-------|----------------|---|--|--------|
> | Base Model |	75 | 0 | 0	|  0 |
> | RL Change |	-20 | 0 | 0  |  0 |

---

> > ### Comment · Reviewer_fv3V · 2025-08-03
> >
> > I appreciate the authors' detailed response. I think my score is appropriate, and I will maintain my positive attitude towards this submission.

---

### Official Review · Reviewer_StAU · 2025-07-02

**Rating:** 5
**Confidence:** 4

**Summary:**

The paper "MathOOD: Probing the Generalization Limits of LLMs in Math Reasoning" introduces a novel benchmark, MathOOD, designed to systematically evaluate the out-of-distribution (OOD) generalization capabilities of large language models (LLMs) in mathematical reasoning. The benchmark focuses on three axes of generalization inspired by Boden’s typology of creativity: exploratory, compositional, and transformative. The authors construct matched training-test pairs across various mathematical domains, including geometry, number theory, algebra, combinatorics, logic, and puzzles. They evaluate several state-of-the-art LLMs and observe significant performance degradation as problem complexity increases. The paper demonstrates that while reinforcement learning (RL) fine-tuning improves exploratory generalization, it has limited impact on compositional and transformative reasoning.

**Dataset Code Accessibility:**

Yes

**Dataset Code Comments:**

This public repo provides code for generating problems by difficulty level.

**Ethical Considerations:**

No, there are no or only very minor ethics concerns

**Limitations Weaknesses:**

L1: The paper does not report error bars or other statistical significance measures for the experimental results. This makes it difficult to assess the reliability and variability of the performance metrics reported.

L2: While the paper focuses on three axes of generalization, it acknowledges that RL fine-tuning has limited impact on compositional and transformative reasoning. This suggests that the benchmark may not fully capture the full spectrum of creative reasoning required for advanced mathematical problem-solving.

L3: The paper mentions the computational resources required for the experiments but does not provide detailed information on the exact hardware and time required for each experimental run. This information is crucial for reproducibility and practical application.

L4: The paper discusses potential positive impacts but could benefit from a more detailed discussion on potential negative societal impacts, such as the misuse of models in educational settings or the implications of biased model performance.

**Strengths Contributions:**

S1: This paper proposes a novel benchmark and conceptual framework. The introduction of MathOOD provides a controlled yet diverse benchmark for evaluating OOD generalization in mathematical reasoning. This is a significant contribution as existing benchmarks either lack the necessary diversity or are too coarse for detailed analysis. In addition, the paper proposes a novel taxonomy of generalization—exploratory, compositional, and transformative—which offers a systematic framework for analyzing mathematical reasoning beyond existing benchmarks.

S2: This paper provides empirical insights. Regarding performance degradation, the study empirically demonstrates that state-of-the-art LLMs, including DeepSeek-R1, Claude 3.7 Sonnet, and OpenAI O4-mini, struggle with increasing problem complexity. This highlights the limitations of current models in handling more complex mathematical problems. Regarding the impact of RL Fine-Tuning, the paper provides valuable insights into the impact of RL fine-tuning on different generalization axes. While RL improves in-distribution and exploratory generalization, it has limited impact on compositional and transformative tasks.

S3: This paper presents controlled synthesis and problem generation. The use of templated problem generators ensures precise control over the diversity, complexity, and specific reasoning strategies required for solutions. This allows for systematic study of generalization and construction of compound examples by fusing multiple templates. The benchmark spans a wide range of mathematical domains, enabling comprehensive evaluation across diverse types of mathematical reasoning and generalization.

S4: The authors provide open access to their code and dataset via a GitHub repository, ensuring reproducibility and enabling further research in this area.

---

> ### Author Rebuttal · Authors · 2025-07-31
>
> **Opening Remark**: We sincerely thank the reviewer for the constructive and valuable feedback. We are honored that the reviewer recognized MathOOD as a significant contribution, highlighting the value of our novel benchmark and conceptual framework. We are equally pleased that our empirical insights into model performance and RL fine-tuning were appreciated, underscoring current limitations and opportunities in mathematical reasoning. Additionally, we are gratified by the recognition of our controlled synthesis approach and commitment to reproducibility through open access to our code and datasets. We address the reviewer’s comments and concerns in detail below.
>
> > **Inclusion of Error Bars and Statistical Significance**
>
> Thank you for the suggestion. We have incorporated error bars indicating the standard deviation (calculated over the last five evaluation checkpoints) into Figure 4. The detailed results with standard deviations are presented below:
>
> |                      | Algebra | Arithmetic | Combinatory | Geometry | Logic | Numbertheory |
> |----------------------|---------|------------|-------------|----------|-------|---------------|
> | ID Improvement       | +27 (±3)    | +37 (±4)   | +60   (±2)    | +18   (±4)  | +52 (±3)  | +33  (±4)      |
> | OOD Improvement      | +16 (±1) | +24  (±6)   | +15  (±2)    | +5  (±0)   | +39  (±3) | +14     (±3)      |
>
> |                         | Comp. Setting 1 | Comp. Setting 2 | Comp. Setting 3 | Comp. Setting 4 | Comp. Setting 5 |
> |-------------------------|----------------|------------------|------------------|------------------|------------------|
> | ID1 Improvement         | +28    (±3)     | +16   (±2)       | +54    (±3)        | +54   (±2)         | +56  (±3)         |
> | ID2 Improvement         | +32     (±2)     | +69    (±4)       | +49   (±2)        | +48  (±4)        | +56  (±3)         |
> | OOD Improvement         | 0   (±0)        | +15      (±5)        | 0    (±0)          | +6    (±1)           | +8  (±0)          |
>
> |                         | Trans. Setting 1 | Trans. Setting 2 |	Trans. Setting 3 | Trans. Setting 4 |
> |-------------------------|----------------|------------------|------------------|------------------|
> | RL Improve (ID) |	+56	(±3) | +39 (±2)| +36 (±4)	| +58 (±3) |
> | RL Change (OOD) |	-30 (±11) | 0 (±0)  | 0 (±0) |  0 (±0) |
>
>
> > **Coverage and Scope of Creative Reasoning**
>
> The three generalization axes analyzed in our study are inspired by Margaret Boden’s typology of creativity: Combinational, Exploratory, and Transformational Creativity. These dimensions primarily reflect creativity as observed in human cognitive behavior. However, we acknowledge the possibility that LLMs might exhibit forms of creativity distinct from human cognition, which are not yet captured by current cognitive science frameworks.
>
>
> > **Detailed Computational Resources and Experimental Setup**
>
> Thank you for highlighting this important aspect. We have provided comprehensive details about our computational resources and experimental setups in Appendix B. Briefly, each reinforcement learning (RL) training run employs 32 NVIDIA H100 GPUs, distributed over 4 computing nodes, and typically completes within approximately 12 hours.
>
>
> > **Consideration of Potential Negative Societal Impacts**
>
> We appreciate this comment and will incorporate the following discussion into our conclusion:
>
> "While MathOOD provides structured training problems that can help improve the mathematical reasoning capabilities of LLMs, several risks must be considered. In educational settings, excessive reliance on these models—even in scenarios where they provide erroneous or biased outputs—could negatively impact learning outcomes or foster unwarranted trust. Furthermore, since MathOOD focuses on specific problem domains, models may exhibit disproportionately stronger performance on certain topics, potentially resulting in an incomplete or skewed perception of their overall mathematical proficiency."

---

### Official Review · Reviewer_PDA1 · 2025-07-03

**Rating:** 5
**Confidence:** 3

**Summary:**

Aiming to evaluate LLMs' out-of-distribution generalization ability on math reasoning, this paper proposes a new benchmark, MathOOD. The problems in the dataset are generated by templated problem generators and calibrated at the knowledge level of AIME. By building different pairs of training and test datasets, the benchmark considers three different generalizations: (1) exploratory generalization (applying known skills to more complex problems), (2) compositional generalization (combing known skills), and (3) transformative generalization (moving beyond known skills and adopting unconventional strategies). The experiments on top-level LLMs indicate notable improvement in exploratory generalization, limited improvement in compositional generalization, and zero improvement in transformativey generalization.

**Dataset Code Accessibility:**

Yes

**Ethical Considerations:**

No, there are no or only very minor ethics concerns

**Final Justification:**

Thanks for the author's responses. I would like to maintain the current positive score.

**Limitations Weaknesses:**

I would like to know the answer to these questions:
1. In Figure 3, the performances of some models on some domains do not monotonically decrease as difficulty increases, such as Claude and two OpenAI models on number theory. Does it mean that the difficulty measure in these domains may not make a lot of sense?
2. For compositional generalization and transformative generalization, could you introduce more about how to create test sets? Is any automatic tool (such as LLMs) used in this pipeline?
3. How to measure the difficulty of test problems for compositional generalization and transformative generalization?

**Strengths Contributions:**

1. The design of three categories of generalization is comprehensive and novel.
2. The program-based templated problem generators can help the research community to build larger datasets in the future.

---

> ### Author Rebuttal · Authors · 2025-07-31
>
> **Opening Remark**: We thank the reviewer for the constructive and valuable feedback. We are honored that the reviewer recognized our design of three generalization categories as comprehensive and novel. Additionally, we appreciate the acknowledgment that our program-based templated problem generators can aid the research community in constructing larger datasets. We have carefully addressed the reviewers' comments and questions below.
>
> > **Non-monotonic Performance Trends with Increasing Difficulty**
>
> Thank you for raising this important point! Our complexity control follows a straightforward and intuitive approach. Specifically, in number theory problems, we define difficulty based on the digit count of the final answer; larger answers typically necessitate additional rounds of reasoning, including prime number identification.
>
> For instance, consider the following problem:
>
> *"Let $p$ be the smallest prime number for which there exists a positive integer $n$ such that $n^3 + 2$ is divisible by $p^4$. Find the smallest positive integer $m$ such that $m^3 + 2$ is divisible by $p^4$."*
>
> Since the answer to this problem is 322, it is classified as difficulty level 3 (3 digits).
>
> The observed non-monotonicity arises due to two factors:
>
> 1. **Sampling Noise**: Due to API budget constraints, we evaluate each difficulty level using only 100 sampled problems, with a single pass\@1 attempt per problem. This limited sampling inherently introduces variability.
>
> 2. **Intrinsic Variability**: Occasionally, a specific problem instance at a given difficulty level may be inherently simpler than certain lower-difficulty instances, depending on parameter values.
>
> Nevertheless, at a distributional level, we ensure that the difficulty increases systematically, as evidenced by the overall downward performance trend in Figure 3.
>
>
> > **Curation Process for Compositional and Transformative Problems**
>
> We intentionally avoided using LLMs in the curation process for compositional and transformative problem sets. This decision was made to prevent trivial or overly familiar problem combinations that LLMs might generate from memory or shallow reasoning.
>
> Instead, our methodology involves:
>
> - Starting with "seed problems" curated by three co-authors experienced in mathematical competitions (e.g., AIME).
> - Perturbing parameters systematically for a larger test set while ensuring genuine reasoning challenges.
>
> This human-driven approach helps maintain the novelty and rigor necessary for meaningful evaluation.
>
> > **Measuring Difficulty in Compositional and Transformative Problems**
>
> Great questions! We conducted comparative analyses of frontier LLMs' performances as detailed in Figure 5 (Appendix). On average, the difficulty level of problems in these two generalization settings aligns approximately with level 3 difficulty, as the middle bar of the subfigures in Figure 3.

---

### Official Review · Reviewer_rFrB · 2025-07-08

**Rating:** 5
**Confidence:** 4

**Summary:**

For resources, the submission

1. designs a math problem generation program, comprising 40 templates spanning 6 mathematical domains;
2. introduces MathOOD, a benchmark utilizing this program to evaluate different axes of generalization.

For methodology, the submission

1. divides generalization into 3 axes: Exploratory, Compositional and Transformative;
2. utilizes MathOOD to evaluate these axes of generalization by training and testing on problems of different templates;
3. evaluates top-tier LLMs and observes sharp performance degradation as problem complexity increases, showing the difficulty of generated problems;
4. evaluates Qwen-series models trained with RL, suggesting that RL can gain notable improvements in exploratory generalization, while compositional generalization remains limited and transformative generalization shows 0 improvement.

**Dataset Code Accessibility:**

Yes

**Ethical Considerations:**

No, there are no or only very minor ethics concerns

**Limitations Weaknesses:**

1. The compositional generalization and transformative generalization rely on relationship between problem templates, which still needs significant manual efforts.
2. Realistic training data usually span across diverse problem types, which might emerge generalization behaviors. The controlled experiments might not be enough to reflect the realistic generalization mechanisms.

**Strengths Contributions:**

1. Generally, the paper is well-written, organized and easy to understand, and the figures, tables, and captions are informative.
2. The submission curates 40 math problem templates across 6 domains, which is a valuable resource.
3. The submission suggests RL's limitation in compositional generalization and transformative generalization, which is interesting.

---

> ### Author Rebuttal · Authors · 2025-07-31
>
> **Opening Remark**: We thank the reviewer for the constructive and valuable feedback. We are honored that the reviewer found our paper "well-written, organized, and easy to understand," and considered our curated collection to be a valuable resource. Additionally, we are pleased that our discussion on the limitations of RL was found interesting. We have carefully addressed the reviewer's comments and questions below.
>
>
> > **Required Manual Efforts in Curating Compositional and Transformative Generalization Problems**
>
> Indeed, generating compositional and transformative generalization problems currently necessitates human effort due to the intrinsic complexity of combining distinct reasoning strategies and creating genuine cognitive leaps. In our compositional setting, the process involves non-trivial insights to meaningfully integrate strategies from two problem domains, rather than superficially concatenating problems. Similarly, transformative generalization inherently requires original "creative insights" to formulate novel, unconventional problem-solving approaches. At present, attempts to automate this with current LLMs yield suboptimal results: compositional problems tend to be shallow concatenations, and transformative problems often rely on known, pre-trained insights rather than novel leaps. While we agree that automation remains a significant challenge, we believe our initial efforts provide a valuable starting point and will stimulate further methodological advancements in this area.
>
> > **Evaluating Generalization under Diverse Problem Settings**
>
> This is an insightful suggestion. To address this, we conducted additional experiments by training on a combined dataset that includes all problem families presented in our study. Specifically, we evaluated whether this diverse training setup could enhance compositional and transformative generalization compared to isolated training. Our results indicate that mixed training across diverse problem families yielded performance improvements in certain compositional settings (particularly Compositional Setting 3 and Setting 5), likely due to beneficial reasoning strategies transferred across domains. However, even with increased training diversity, transformative generalization saw negligible improvement, underscoring its inherent difficulty and highlighting an important limitation for future exploration.
>
> |     | Comp. Setting 1 | Comp. Setting 2 | Comp. Setting 3 | Comp. Setting 4 | Comp. Setting 5 |
> |-------------------------|----------------|------------------|------------------|------------------|------------------|
> | RL Improvement on OOD (Isolated Training - Figure 4c)        | 0      | +15     | 0       | +6        | +8   |
> | RL Improvement on OOD (Mixed Training on All Problem Families)       | 0      | +14     | +5       | +6        | +14   |

---

### Comment · Area_Chair_QJSs · 2025-08-04
**Author-Reviewer Discussions**

Dear Reviewers,

Thank you for your time and valuable feedback on this submission. The authors have submitted their responses to your comments and suggestions.

- If your concerns have been sufficiently addressed in the authors' response, we kindly ask you to update your rating accordingly.

- If you require further clarification or have additional questions for the authors, please submit them as soon as possible **before Aug 6**, which will allow the authors adequate time to respond.

Best regards,

AC

---

### Decision · Program_Chairs · 2025-09-18

**Decision:**

Accept (poster)

**Comment:**

This paper introduces MathOOD, a new benchmark for evaluating the out-of-distribution (OOD) generalization of Large Language Models (LLMs) in mathematical reasoning. MathOOD uses templated problem generators to create AIME-level difficulty problems. It tests three generalization types: exploratory (applying known skills to harder problems), compositional (combining known skills), and transformative (using novel strategies). Experiments on top LLMs show notable gains only in exploratory generalization, with limited to no improvement in compositional and transformative generalization.

All reviewers agree that this submission is well-written, presents a comprehensive benchmark (with 40 math problem templates across 6 domains), and provides valuable empirical insights. The authors' response has adequately addressed all previous concerns. Consequently, all reviewers recommend acceptance. This submission is well above the bar for the NeurIPS DB track and should be accepted.